# VGLL2 and TEAD1 fusion proteins identified in human sarcoma drive YAP/TAZ-independent tumorigenesis by engaging EP300

Susu Guo[1†], Xiaodi Hu[2†], Jennifer L Cotton[2], Lifang Ma[1], Qi Li[2,3], Jiangtao Cui[1], Yongjie Wang[1], Ritesh P Thakare[2], Zhipeng Tao[4], Y Tony Ip[3], Xu Wu[4], Jiayi Wang[1]*, Junhao Mao[2]*

[1]Department of Clinical Laboratory, Shanghai Chest Hospital, Shanghai Jiao Tong University School of Medicine, Shanghai, China; [2]Department of Molecular, Cell and Cancer Biology, University of Massachusetts Chan Medical School, Worcester, United States; [3]Program in Molecular Medicine, University of Massachusetts Chan Medical School, Worcester, United States; [4]Cutaneous Biology Research Center, Massachusetts General Hospital, Harvard Medical School, Charlestown, United States

*For correspondence:
jiayi.wang@sjtu.edu.cn (JW);
Junhao.Mao@umassmed.edu
(JM)

[†]These authors contributed
equally to this work

Competing interest: The authors
declare that no competing
interests exist.

Reviewing Editor: Xiaobing
Shi, Van Andel Institute, United
States

## eLife Assessment

This is a **valuable** study describing how rhabdomyosarcoma fusion-oncogenes, VGLL2-NCOA2 and TEAD1-NCOA2, function at the genomic, transcriptional, and proteomic levels in multiple systems. The experimental data is **convincing**, supporting a model in which these fusion-oncogenes leverage TEAD transcriptional signatures independent of YAP/TAZ. This work offers new mechanistic insights into oncogenic gene fusion events and reveals potential therapeutic strategies for the treatment of rhabdomyosarcomas.

**Abstract** Studies on Hippo pathway regulation of tumorigenesis largely center on YAP and TAZ, the transcriptional co-regulators of TEADs. Here, we present an oncogenic mechanism involving VGLL and TEAD fusions that is Hippo pathway-related but YAP/TAZ-independent. We characterize two recurrent fusions, *VGLL2-NCOA2* and *TEAD1-NCOA2*, recently identified in human spindle cell rhabdomyosarcoma. We demonstrate that in contrast to VGLL2 and TEAD1 the fusion proteins are potent activators of TEAD-dependent transcription, and the function of these fusion proteins does not require YAP/TAZ. Furthermore, we identify that VGLL2 and TEAD1 fusions engage specific epigenetic regulation by recruiting histone acetyltransferase EP300 to control TEAD-mediated transcriptional and epigenetic landscapes. We show that small-molecule EP300 inhibition can suppress fusion protein-induced oncogenic transformation both in vitro and in vivo in mouse models. Overall, our study reveals a molecular basis for VGLL involvement in cancer and provides a framework for targeting tumors carrying *VGLL*, *TEAD*, or *NCOA* translocations.

## Introduction

Hippo signaling, originally identified in *Drosophila* as an organ size control pathway, has emerged as a critical developmental pathway whose dysregulation contributes to the development and progression of a variety of human diseases, including cancer (*Zheng and Pan, 2019*; *Ma et al., 2019*; *Calses*

*et al., 2019*; *Kulkarni et al., 2020*; *Piccolo et al., 2023*). In the mammalian Hippo pathway, activation of the core kinase cascade, which comprises the macrophage stimulating 1/2 (MST1/2) and large tumor suppressor kinase 1/2 (LATS1/2) kinases, leads to phosphorylation, cytosolic retention, and degradation of the key transcriptional coactivators, YAP and TAZ. Upon Hippo pathway inactivation, YAP and TAZ translocate into the nucleus and interact with the TEAD family of transcription factors, thereby inducing downstream gene transcription (*Zheng and Pan, 2019*; *Ma et al., 2019*; *Totaro et al., 2018*). Although the Hippo pathway has been implicated in human cancers, mutations or deletions of the pathway components, such as the MST1/2 or LATS1/2 kinases, are rarely detected in cancers. The focus on Hippo involvement in tumorigenesis has been on the transcriptional co-activators, YAP and TAZ, whose upregulation or nuclear accumulation has been reported in various cancer types (*Piccolo et al., 2023*; *Thompson, 2020*; *Franklin et al., 2023*). In addition, recent studies identified YAP and TAZ fusion proteins as oncogenic drivers in several tumors, including epithelioid hemangioendothelioma, supratentorial ependymoma, porocarcinoma, epithelioid fibrosarcoma, and NF2-wildtype meningioma (*Driskill et al., 2021*; *Seavey et al., 2021*; *Merritt et al., 2021*; *Szulzewsky et al., 2021*; *Szulzewsky et al., 2022*; *Garcia et al., 2022*), further highlighting the critical role of YAP/TAZ in tumorigenesis.

The mammalian VGLL family proteins (VGLL1-4) are expressed in various tissues, and it is thought that VGLL proteins carry out their cellular function via interaction with TEADs (*Faucheux et al., 2010*; *Pobbati et al., 2012*; *Simon et al., 2016*; *Yamaguchi, 2020*). However, in comparison to YAP and TAZ, little is known about their functional regulation. Among the VGLL proteins, VGLL4 and its *Drosophila* homolog Tgi have been demonstrated as transcriptional repressors by competing with YAP/TAZ binding to TEADs through two TONDU (TDU) domains (*Koontz et al., 2013*; *Cai et al., 2022*; *Zhang et al., 2014*; *Li et al., 2023*). In contrast, the precise function of VGLL1-3 in both normal and tumor cells and how they influence Hippo/YAP signaling and affect TEAD transcriptional output remain poorly understood.

Several recent studies have identified the recurrent rearrangement of the *VGLL2* and *TEAD1* genes in a large subset of spindle cell rhabdomyosarcoma (scRMS), a pediatric form of RMS that is distinct from embryonic RMS (ERMS) (*Alaggio et al., 2016*; *Tan et al., 2020*; *Chen et al., 2020*; *Cyrta et al., 2021*; *Whittle et al., 2022*). In these fusions, *VGLL2* and *TEAD1* participate as 5′ partners in the recurrent translocations, and their 3′ fusion partners often involve the *NCOA2* gene on 8q13.3 (*Alaggio et al., 2016*). Among these fusion proteins, the *VGLL2-NCOA2* fusion has recently been shown to drive tumor formation in zebrafish and allograft models (*Watson et al., 2023*). Here, we explored the molecular mechanism underlying the oncogenic transformation of VGLL2-NCOA2 and TEAD1-NCOA2 fusion proteins. We revealed that, in comparison to VGLL2 and TEAD1, VGLL2-NCOA2 and TEAD1-NCOA2 are robust activators of TEAD-mediated transcription. We demonstrated that VGLL2-NCOA2 and TEAD1-NCOA2-controlled transcriptional activation is YAP/TAZ-independent, and these fusion proteins specifically engage the CREBBP/EP300 epigenetic factors that are critical for their oncogenic activation.

## Results

### VGLL2-NCOA2 and TEAD1-NCOA2 induce TEAD-mediated transcriptional activation

In *VGLL2-NCOA2* and *TEAD1-NCOA2* fusions, *VGLL2* and *TEAD1* are 5′ partners and maintain the key functional domains at the N-terminus. *VGLL2* retains the TDU domain, while *TEAD1* preserves its TEA DNA binding domain (*Figure 1A*). The 3′ partners of the fusions are *NCOA2*, which retains its two C-terminal transcriptional activation domains (TAD) (*Figure 1A*). The exons and breaking points of the *VGLL2*, *TEAD1*, and *NCOA2* genes involved in generating the *VGLL2-NCOA2* and *TEAD1-NCOA2* gene arrangement are presented in *Figure 1—figure supplement 1A*. To test the transcriptional activity of the fusion proteins, we utilized an 8✕GIIC luciferase reporter that contains 8✕TEAD DNA binding sites (TBS-Luc) (*Dupont et al., 2011*). When ectopically expressed in HEK293T cells, we found that, in comparison to the VGLL2, TEAD1, and NCOA2 proteins, the VGLL2-NCOA2 and TEAD1-NCOA2 fusion proteins could significantly induce the transcriptional activation of the TBS-Luc reporter, and their activity was comparable to the activated forms of YAP and TAZ, YAP$^{5SA}$ and TAZ$^{4SA}$, in which the LATS kinase phosphorylation sites are mutated rendering them constitutively active (*Zhao*

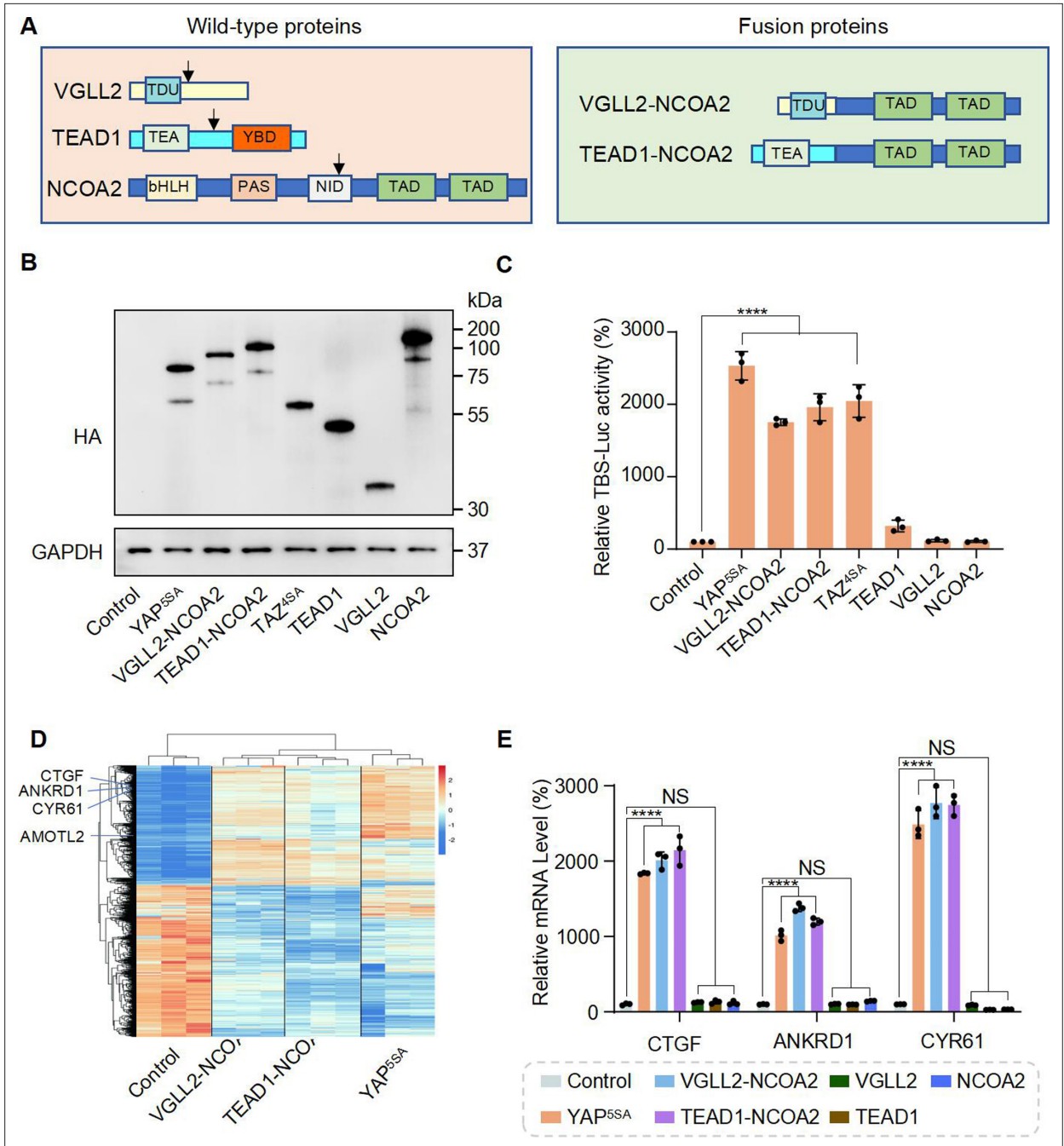

**Figure 1.** VGLL2-NCOA2 and TEAD1-NCOA2 Induce TEAD-mediated transcriptional activation. (**A**) Schematic representation of protein structure of VGLL2, TEAD1, NCOA2, VGLL2-NCOA2, and TEAD1-NCOA2. Tondu motif (TDU), TEA DNA binding domain (TEA), YAP binding domain (YBD), basic Helix-Loop-Helix (bHLH), Per-Arnt-Sim domain (PAS), nuclear receptor interaction domain (NID), and transcriptional activation domain (TAD). Arrows point to the break points. (**B**) Immunoblot analysis of YAP5SA, VGLL2-NCOA2, TEAD1-NCOA2, TAZ4SA, TEAD1, VGLL2, and NCOA2 in HEK293T cells transfected with the expression constructs carrying the HA tag. The figure shows the representative results of three biological replicates. (**C**) YAP5SA, VGLL2-NCOA2, TEAD1-NCOA2, TAZ4SA, TEAD1, VGLL2, and NCOA2 induce transcriptional activation of TBS (TEAD-binding site)-luciferase reporter (TBS-Luc) in HEK293T cells. Data were expressed as mean ± SD. n=3. ****p<0.0001. (**D**) Heatmap showing expression levels of the core genes including *CCN2, CCN1, ANKRD1,* and *AMOTL2* significantly regulated in HEK293T cells expressing YAP5SA, VGLL2-NCOA2, and TEAD1-NCOA2. N=3. (**E**) mRNA expression levels of *CCN2, ANKRD1,* and *CCN1* in HEK293T cells expressing YAP5SA, VGLL2-NCOA2, TEAD1-NCOA2, TEAD1, VGLL2, or NCOA2. Data were expressed as mean ± SD. n=3; ****p<0.0001. NS, no significance.

The online version of this article includes the following source data and figure supplement(s) for figure 1:

**Source data 1.** Original western blot membranes corresponding to *Figure 1B* indicating the relevant bands.

*Figure 1 continued on next page*

*Figure 1 continued*

**Source data 2.** Original western blot membranes corresponding to *Figure 1B* indicating the relevant bands.

**Figure supplement 1.** VGLL2-NCOA2 regulates TEAD-dependent reporter activity.

**Figure supplement 1—source data 1.** Original western blot membranes corresponding to *Figure 1—figure supplement 1C and E* indicating the relevant bands.

**Figure supplement 1—source data 2.** Original western blot membranes corresponding to *Figure 1—figure supplement 1C and E* indicating the relevant bands.

**Figure supplement 2.** Analysis of VGLL2-NCOA2, TEAD1-NCOA2, and YAP$^{5SA}$ -induced transcriptomes.

---

*et al., 2010*; *Figure 1B and C*). Additionally, we showed that ectopic expression of VGLL2-NCOA2 and TEAD1-NCOA2 in HEK293T cells promotes cell proliferation (*Figure 1—figure supplement 1B*), mimicking the pro-growth activity of activated YAP and TAZ. Furthermore, we demonstrated that the ability of VGLL2-NCOA2 to induce downstream transcription requires both its N-terminal VGLL2 and C-terminal NCOA2 fusion parts, and this effect is dose-dependent and not affected by the 5′ or 3′ protein tags (*Figure 1—figure supplement 1C–H*).

To further characterize the downstream transcription induced by VGLL2-NCOA2 and TEAD1-NCOA2, we performed RNA-seq analysis in HEK293T cells overexpressing VGLL2-NCOA2, TEAD1-NCOA2, or YAP$^{5SA}$ (*Figure 1D*, *Figure 1—figure supplement 2*, *Supplementary files 1–3*). We found that all three could robustly drive transcriptional programs in HEK293T cells, and VGLL2-NCOA2- and TEAD1-NCOA2-induced transcription profiles clustered together and appeared to be distinct from the YAP$^{5SA}$-driven program (*Figure 1D*). However, similar to activated YAP, the VGLL2-NCOA2, and TEAD1-NCOA2 fusion proteins also strongly induced the expression of *bona fide* TEAD downstream target genes, such as *CCN2 (CTGF)*, *CCN1 (CYR61)*, *ANKRD1*, and *AMOTL2* (*Figure 1D*). This was further confirmed by RT-qPCR analysis, which showed that the transcriptional levels of *CCN1*, *CCN2*, and *ANKRD1* were significantly elevated in the cells expressing VGLL2-NCOA2, TEAD1-NCOA2, or YAP$^{5SA}$ but not in those expressing VGLL2, TEAD1, and NCOA2 (*Figure 1E*). Taken together, these data suggest that unlike native VGLL2 and TEAD1, VGLL2-NCOA2 and TEAD1-NCOA2 fusion proteins function as strong activators of TEAD-mediated transcription.

## VGLL2-NCOA2 and TEAD1-NCOA2-induced transcription does not require YAP and TAZ

Next, we set out to examine how the VGLL2-NCOA2 and TEAD1-NCOA2 fusion proteins interact with YAP/TAZ and TEAD, the well-characterized transcriptional nexus in the Hippo pathway. We performed co-IP assays in HEK293T cells expressing VGLL2-NCOA2, YAP$^{5SA}$, and TEAD1 and demonstrated that VGLL2-NCOA2 was strongly associated with TEAD1 but not YAP$^{5SA}$ (*Figure 2A*). Furthermore, we showed that VGLL2-NCOA2 could bind to the endogenous TEAD proteins; however, we did not detect a strong interaction between VGLL2-NCOA2 and endogenous YAP and TAZ proteins in HEK293T cells (*Figure 2B*). In addition, we compared the ability of TEAD1-NCOA2 and TEAD1 to interact with YAP and TAZ. Not surprisingly, TEAD1 was able to bind YAP/TAZ; however, the interaction between TEAD1-NCOA2 and endogenous YAP/TAZ was not detected (*Figure 2C*), which is consistent with the absence of the C-terminal YAP/TAZ binding domain in the TEAD1-NCOA2 fusion. Together, these co-IP assays revealed the lack of association between the VGLL2 and TEAD1 fusion proteins and YAP/TAZ, suggesting that the regulation of transcription by the fusion proteins is likely TEAD-dependent but YAP/TAZ-independent.

To examine the TEAD regulation in VGLL2-NCOA2 and TEAD1-NCOA2-mediated transcription, we first utilized a small-molecule TEAD inhibitor, CP1, that was recently developed to inhibit TEAD auto-palmitoylation, leading to protein instability and transcriptional inactivation (*Li et al., 2020*; *Sun et al., 2022*). Upon treatment with the TEAD palmitoylation inhibitor CP1 in HEK293T cells transfected with YAP$^{5SA}$, VGLL2-NCOA2, and TEAD1-NCOA2, we found that CP1 effectively inhibited YAP$^{5SA}$- and VGLL2-NCOA2-induced transcriptional activation but had largely no effect on TEAD1-NCOA2-induced reporter activation (*Figure 2D*). The inability of CP1 to inhibit TEAD1-NCOA2 activity is likely because the TEAD1-NCOA2 fusion does not contain the auto-palmitoylation sites in the C-terminal YAP-binding domain of TEAD1. To further explore TEAD dependence, we generated a TEAD fusion-dominant repressor, TEAD-ENR, which fuses its N-terminal TEA DNA-binding domain to a C-terminal

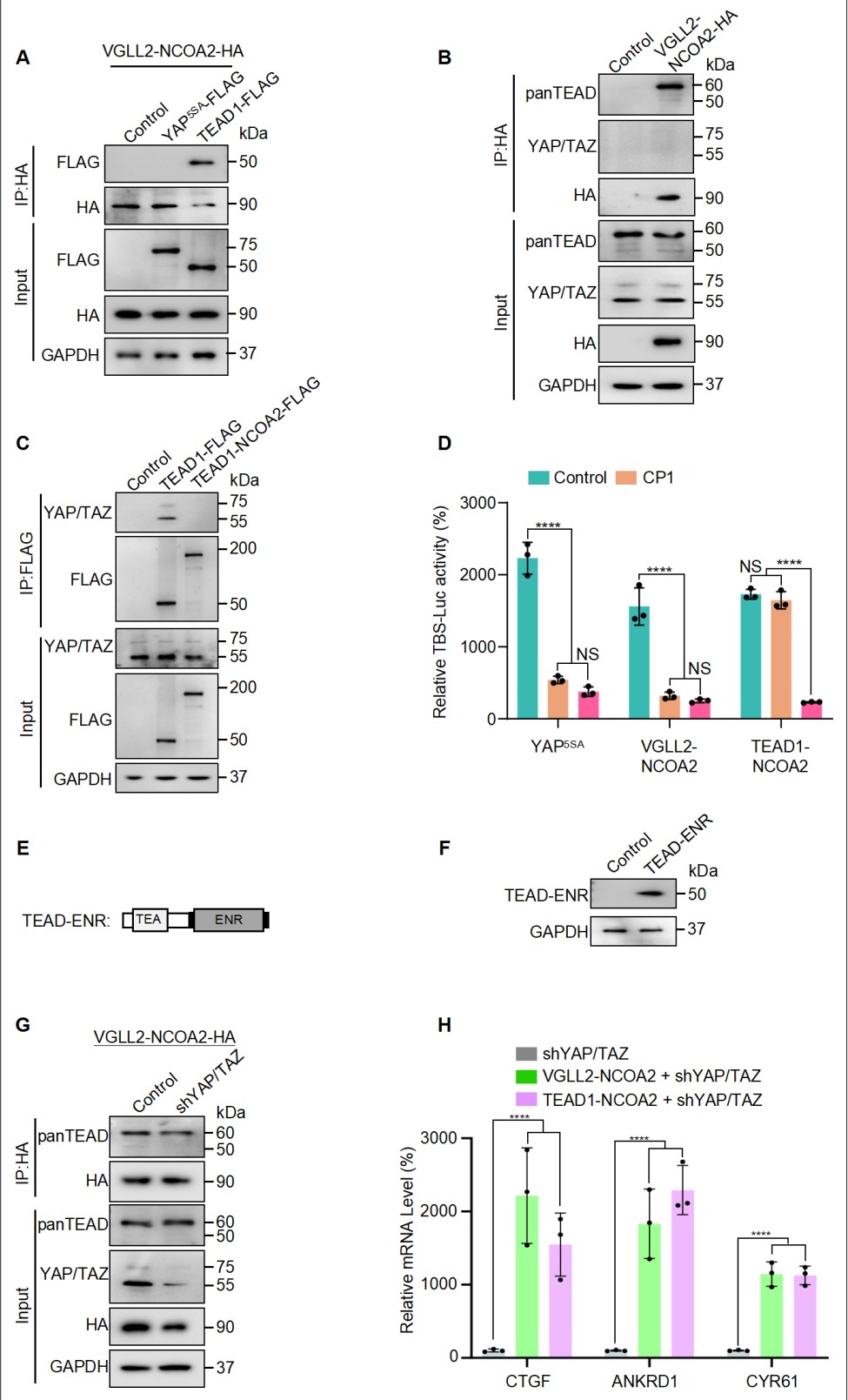

**Figure 2.** VGLL2-NCOA2 and TEAD1-NCOA2-induced transcription does not require YAP and TAZ. (**A**) Co-IP assays showing VGLL2-NCOA2 binding to TEAD1 but not YAP[5SA]. YAP[5SA]-Flag or TEAD1-Flag was co-expressed with VGLL2-NCOA2-HA in HEK293T cells and immunoprecipitated with an anti-HA antibody. (**B**) VGLL2-NCOA2 binds to endogenous TEAD but not YAP/TAZ. Endogenous YAP/TAZ and TEAD proteins in HEK293T cells were

*Figure 2 continued on next page*

*Figure 2 continued*

detected by anti-YAP/TAZ and panTEAD antibodies, respectively. (**C**) Co-IP assays showing endogenous YAP/TAZ binding to TEAD1 but not TEAD1-NCOA2. TEAD1-Flag or TEAD1-NCOA2-Flag was expressed in HEK293T cells and immunoprecipitated with an anti-Flag antibody. Endogenous YAP/TAZ proteins were detected by anti-YAP/TAZ antibodies. (**D**) The activity of TBS-Luc reporter in HEK293T cells expressing YAP5SA, VGLL2-NCOA2, or TEAD1-NCOA2, with TEAD inhibitor CP1 (5 μM) treatment or co-expression of TEAD-ENR repressor construct. Data were expressed as mean ± SD. n=3; ****p<0.0001. NS, no significance. (**E**) Schematic representation of TEAD-ENR. TEA DNA-binding domain (TEA) and Engrailed repressor domain (ENR). (**F**) Immunoblot analysis of TEAD-ENR expression in HEK293T cells. (**G**) Co-IP assays showing YAP/TAZ were not essential for VGLL2-NCOA2 binding to endogenous TEADs. VGLL2-NCOA2-HA was expressed in HEK293T cells with or without YAP/TAZ knockdown and immunoprecipitated with an anti-HA antibody. (**H**) Relative mRNA levels of *CCN2*, *ANKRD1*, and *CCN1* in HEK293T cells with YAP/TAZ knockdown and expressing VGLL2-NCOA2 or TEAD1-NCOA2. Data were expressed as mean ± SD. n=3; ****p<0.0001.

The online version of this article includes the following source data for figure 2:

**Source data 1.** Original western blot membranes corresponding to *Figure 2A, B, C, F and G* indicating the relevant bands.

**Source data 2.** Original western blot membranes corresponding to *Figure 2A, B, C, F and G*.

Engrained repressor domain (*Figure 2E and F*). When co-expressed with YAP5SA, VGLL2-NCOA2, and TEAD1-NCOA2 in HEK293T cells, TEAD-ENR potently inhibited the transcriptional activation of all three proteins (*Figure 2D*). These results from both the TEAD inhibitor and transcriptional repressor suggest that downstream TEAD activity and TEAD DNA binding are critical for both fusion proteins.

To further examine the involvement of YAP and TAZ in VGLL2-NCOA2- and TEAD1-NCOA2-induced transcription, we used lentiviral-based shRNA to knock down the expression of YAP and TAZ in HEK293T cells (*Figure 2G*). We showed that VGLL2-NCOA2 could bind to endogenous TEAD proteins at similar levels in both control and YAP/TAZ-knockdown HEK293T cells (*Figure 2G*), suggesting its association with TEADs does not require YAP/TAZ. Furthermore, we found that both VGLL2-NCOA2 and TEAD1-NCOA2 still robustly induced the transcription of the known TEAD target genes, including *CCN1*, *CCN2*, and *ANKRD1*, in HEK293T cells with YAP/TAZ knockdown (*Figure 2H*). Taken together, our data suggest that VGLL2-NCOA2- and TEAD1-NCOA2-induced transcriptional activation requires TEAD DNA binding but is YAP/TAZ independent.

## Characterization of VGLL2-NCOA2- and YAP-controlled transcriptional and chromatin landscapes

To understand the mechanism of how the fusion proteins may control chromatin landscapes and drive downstream transcription, we performed analyses of ATAC-seq, CUT&RUN sequencing, and RNA-seq in HEK293T cells that ectopically expressed VGLL2-NCOA2 and compared them to the epigenetic and transcriptional programs controlled by YAP5SA.

By intersecting the datasets of ATAC-seq (assessing genome-wide chromatin accessibility regulated by VGLL2-NCOA2 and YAP5SA), CUT&RUN (mapping both VGLL2-NCOA2 and YAP5SA binding sites across the genome), and RNA-seq (transcriptome profiling in VGLL2-NCOA2 and YAP5SA-expressing cells), we determined that VGLL2-NCOA2 and YAP controlled overlapping yet distinct chromatin landscapes and downstream transcriptional programs (*Figure 3*, *Figure 3—figure supplement 1*).

In our ATAC-seq assay, the analysis of differentially accessible chromatin sites annotated to the nearest genes showed significant enrichment of Hippo pathway-related genes in both VGLL2-NCOA2 and YAP5SA-expressing cells (*Figure 3A–C*), suggesting that VGLL2-NCOA2 and YAP5SA likely drive open chromatin architecture in a shared set of genes. This is also consistent with our analysis of the genomic occupancy profiles of VGLL2-NCOA2 and YAP5SA defined by CUT&RUN sequencing (*Figure 3D-F*). Annotation of CUT&RUN peaks with respect to their nearest gene transcriptional start site (TSS) showed that both VGLL2-NCOA2 and YAP preferred to occupy distal genomic regions that are not in close proximity to the TSS (0–2 kb) (*Figure 3D*). This is consistent with prior reports that YAP preferentially binds to distal enhancer regions (*Zanconato et al., 2015*; *Liu et al., 2016*) and indicates that the VGLL2-NCOA2 fusion protein also regulates downstream gene transcription through active enhancers.

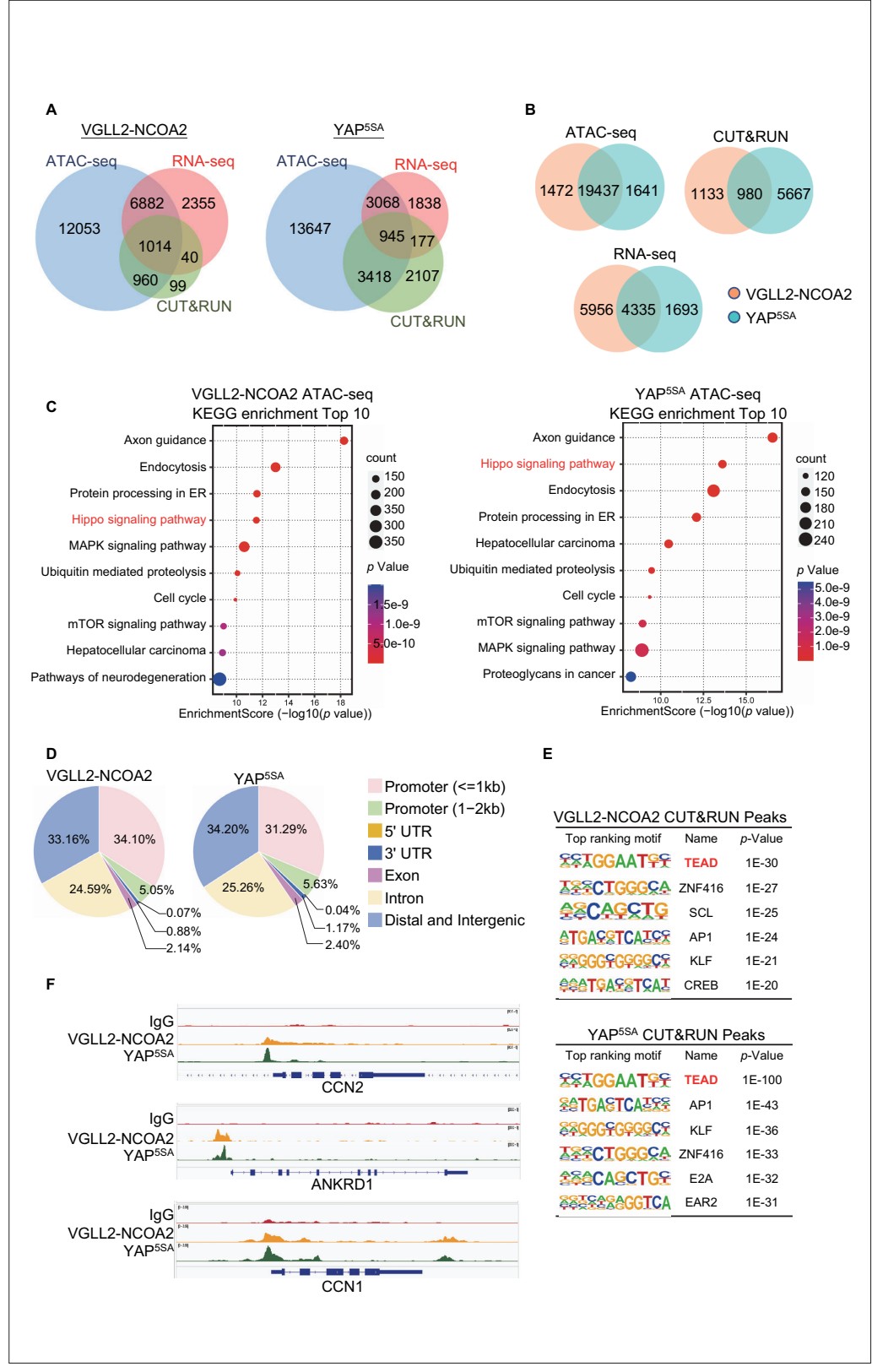

**Figure 3.** Characterization of VGLL2-NCOA2- and YAP-controlled transcriptional and chromatin landscapes. (**A**) Intersection of ATAC-seq (n=2), RNA-seq (n=3), and CUT&RUN (n=2) datasets in HEK293T cells expressing VGLL2-NCOA2 or YAP5SA. (**B**) Venn diagrams showing the overlaps of ATAC-seq peaks, CUT&RUN peaks, and differentially regulated genes from RNA-seq in HEK293T cells expressing VGLL2-NCOA2 or YAP5SA. (**C**) KEGG

*Figure 3 continued on next page*

*Figure 3 continued*

pathway enrichment analysis of ATAC-seq peaks identified in HEK293T cells expressing VGLL2-NCOA2 or YAP[5SA].
The 'Hippo signaling pathway' is highlighted in red. (**D**) Distribution of CUT&RUN binding sites for VGLL2-NCOA2
and YAP[5SA]. (**E**) Motif enrichment analysis of VGLL2-NCOA2 and YAP[5SA] CUT&RUN Peaks. (**F**) Genomic tracks
showing VGLL2-NCOA2 and YAP[5SA] occupancy at the *CCN2, ANKRD1,* and *CCN1* loci.

The online version of this article includes the following figure supplement(s) for figure 3:

**Figure supplement 1.** ATAC-seq and CUT&RUN data characterization in VGLL2-NCOA2 and YAP[5SA]-expressing
cells.

In addition, de novo motif analysis of our CUT&RUN datasets revealed that the TEAD binding
sequence was among the most enriched motifs in both VGLL2-NCOA2 and YAP[5SA] CUT&RUN peaks
(*Figure 3E*), consistent with our observation that TEAD DNA binding was essential for transcriptional activation induced by the fusion proteins (*Figure 2A–F*). Interestingly, the AP1 motif was also
enriched in VGLL2-NCOA2 CUT&RUN peaks (*Figure 3E*), highlighting its proposed roles in TEAD-mediated transcription (*Zanconato et al., 2015*; *Liu et al., 2016*). Furthermore, we compared the
binding profiles of the *bona fide* downstream target genes of VGLL2-NCOA2 and YAP identified in
our CUT&RUN analysis, *CCN1, CCN2,* and *ANKRD1*, and showed that VGLL2-NCOA2 and YAP occupied the same genomic regions in the loci of the three genes (*Figure 3F*). Our analyses demonstrated
that VGLL2-NCOA2 and YAP control overlapping yet distinct chromatin landscapes and occupy a
shared set of genomic regions, thereby driving downstream gene transcription.

## VGLL2-NCOA2 and TEAD1-NCOA2 engage EP300 epigenetic regulators

In addition to their YAP/TAZ independence, the distinct nature of transcriptional and chromatin landscapes controlled by VGLL2-NCOA2 and YAP prompted us to hypothesize that the fusion proteins
might utilize different sets of epigenetic/transcriptional regulators to modify chromatin and drive gene
expression. To this end, we adopted a proximity-labeling proteomics approach called BioID (*Kim and
Roux, 2016*) to identify the potential binding partners of VGLL2-NCOA2 and TEAD1-NCOA2 and
compared them to YAP and TAZ.

In our BioID proteomics screens, we tagged YAP[5SA], TAZ[4SA], VGLL2-NCOA2, and TEAD1-NCOA2
with BirA* (a promiscuous version of the biotin ligase BirA) and used them as baits to probe the
proteomes associated with these proteins (*Figure 4A*, *Supplementary file 4*). By intersecting the
hits associated with YAP/TAZ or VGLL2-NCOA2/TEAD1-NCOA2, we found that TEAD proteins were
among the few proteins associated with all four proteins, providing validation of our BioID datasets
(*Figure 4A*). Focusing on epigenetic and transcriptional regulators identified in the screens, we found
that CREBBP (CBP) and EP300 (P300) were among the hits specifically associated with VGLL2-NCOA2
and TEAD1-NCOA2 in the BioID assays (*Figure 4A*).

The closely related histone acetyltransferases CREBBP and EP300, commonly referred to as
CREBBP/EP300, are known for their function to induce histone H3 lysine 27 acetylation (H3K27ac)
at the promoter and enhancer regions, thereby activating gene transcription (*Ogryzko et al., 1996*;
*Bannister and Kouzarides, 1996*; *Waddell et al., 2021*; *Hogg et al., 2021*). More importantly,
CREBBP/EP300 has been reported to form a complex with the NCOA family proteins that play a critical role in the regulation of nuclear receptors-mediated transcription (*Waddell et al., 2021*; *Yi et al.,
2021*). It is consistent with our proteomics data that the C-terminal NCOA part of VGLL2-NCOA2 and
TEAD1-NCOA2 fusions interacted with CREBBP/EP300 (*Figure 4A*). Interestingly, our motif enrichment analysis identified CREB motif as specifically enriched among VGLL2-NCOA2 CUT&RUN peaks,
but not in YAP[5SA] peaks (*Figure 3E*). CREB is a known transcription factor interacting with CREBBP/
EP300 (*Chrivia et al., 1993*), further supporting the notion that VGLL2-NCOA2 and TEAD1-NCOA2
fusion proteins might recruit CREBBP/EP300. In addition, the activity of CREBBP/EP300 has been
implicated in several types of human cancers (*Waddell et al., 2021*; *Hogg et al., 2021*). Thus, we set
out to further examine the interaction between CREBBP/EP300 and the fusion proteins and how it
may affect downstream transcriptional activation.

We performed the co-IP analysis in HEK293T cells overexpressing VGLL2-NCOA2, TEAD1-NCOA2,
and YAP[5SA] and demonstrated that VGLL2-NCOA2 and TEAD1-NCOA2, but not YAP[5SA], were strongly
associated with endogenous EP300 proteins (*Figure 4B*), suggesting the specific engagement of

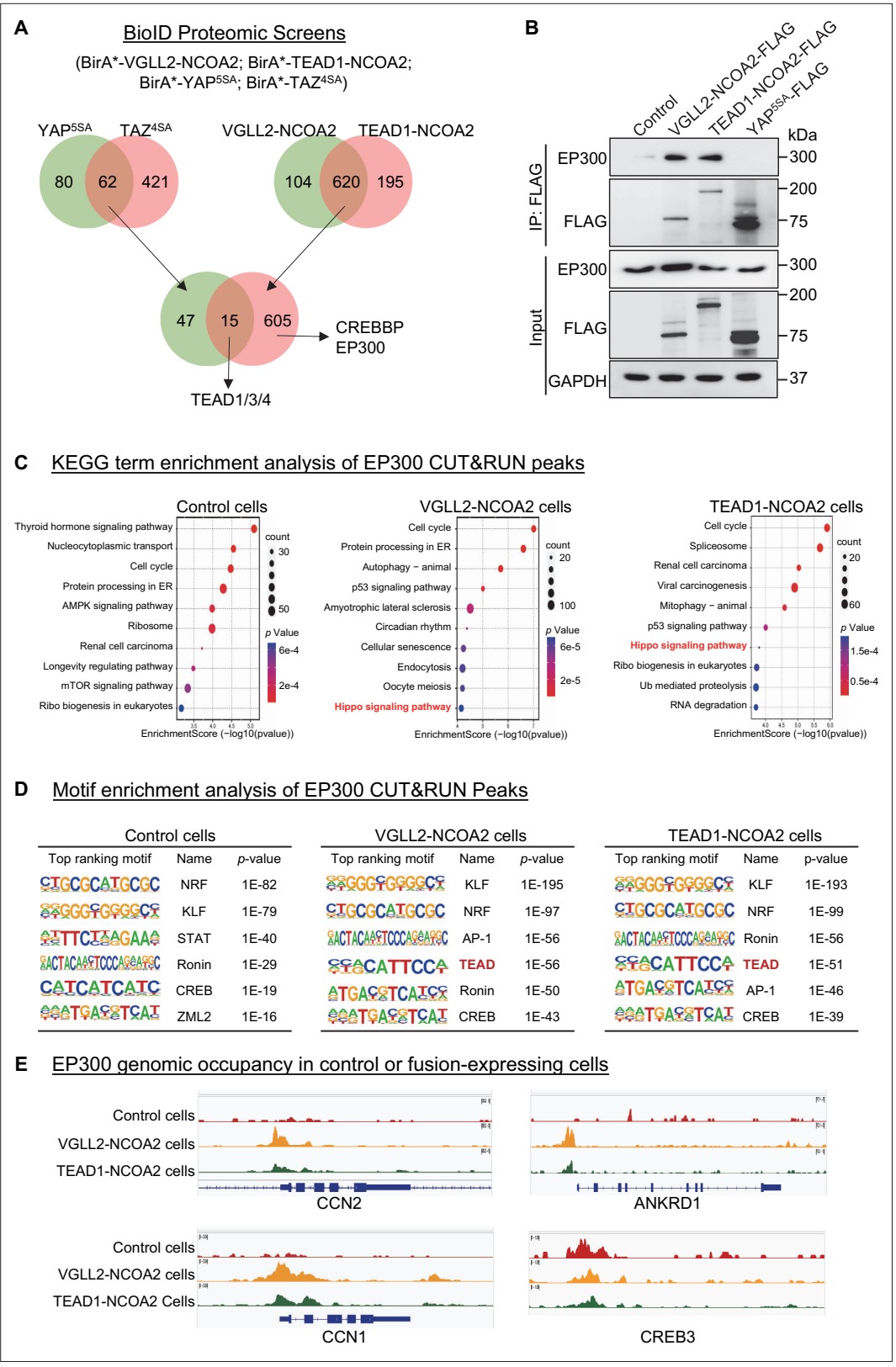

**Figure 4.** VGLL2-NCOA2 and TEAD1-NCOA2 engage EP300 epigenetic regulators. (**A**) Diagram showing the BioID proteomic analyses of BirA*-VGLL2-NCOA2, BirA*-TEAD1-NCOA2, BirA*-YAP^5SA, and BirA*-TAZ^4SA. (**B**) Co-IP assays showing endogenous EP300 binding to VGLL2-NCOA2 and TEAD1-NCOA2 but not YAP^5SA. (**C**) KEGG enrichment analysis of EP300 CUT&RUN peaks in control HEK293T cells and HEK293T cells expressing VGLL2-

*Figure 4 continued on next page*

*Figure 4 continued*

NCOA2 or TEAD1-NCOA2. The 'Hippo signaling pathway' is highlighted in red. (**D**) Motif enrichment analysis of EP300 CUT&RUN peaks in control HEK293T cells and HEK293T cells expressing VGLL2-NCOA2 or TEAD1-NCOA2. (**E**) Genomic tracks showing EP300 occupancy at the *CCN1*, *ANKRD1*, *CCN2*, and *CREB3* loci in control HEK293T cells and HEK293T cells expressing VGLL2-NCOA2 or TEAD1-NCOA2.

The online version of this article includes the following source data for figure 4:

**Source data 1.** Original western blot membranes corresponding to *Figure 4B* indicating the relevant bands.

**Source data 2.** Original western blot membranes corresponding to *Figure 4B* indicating the relevant bands.

EP300 by fusion proteins. To further examine the potential functional regulation of fusion protein-mediated transcription by CREBBP/EP300, we carried out the CUT&RUN sequencing analysis to map EP300 genomic occupancy sites in control and VGLL2-NCOA2- or TEAD1-NCOA2-expressing HEK293T cells (*Figure 4C–E*). Pathway enrichment analysis of EP300 binding peaks annotated by the nearest genes showed significant enrichment of Hippo pathway-related genes in both VGLL2-NCOA2 and TEAD1-NCOA2 expressing cells but not in the control cells (*Figure 4C*), suggesting that VGLL2-NCOA2 and TEAD1-NCOA2 fusion proteins recruit the CREBBP/EP300 complex to a specific set of genes, likely including the TEAD-dependent downstream target genes. In keeping with this result, our de novo motif analysis also revealed that TEAD and AP1 binding sequences were among the top-ranking motifs identified in EP300 occupied genomic sites, specifically in VGLL2-NCOA2- and TEAD1-NCOA2-expressing cells (*Figure 4D*). Interestingly, we noticed that the CREB binding sequence was one of the top-ranking motifs identified in EP300 peaks in both control and VGLL2-NCOA2/TEAD1-NCOA2-expressing cells (*Figure 4D*), consistent with the notion that the CREB family transcription factors are the downstream binding partners of CREBBP/EP300, further validating our CUT&RUN datasets. Upon further examining the binding profiles of EP300 in the loci of *CCN1*, *CCN2*, and *ANKRD1*, we showed that EP300 bound to these loci in VGLL2-NCOA2/TEAD1-NCOA2-expressing cells, but not in control cells (*Figure 4E*). More importantly, EP300 bound to the same genomic regions that were also occupied by VGLL2-NCOA2 identified in VGLL2-NCOA2 CUT&RUN assay (*Figure 3F*). As a control, we showed that EP300 occupied the genomic sites within the *CREB3* locus in cells with or without the expression of the fusion proteins (*Figure 4E*). Taken together, these data suggest that VGLL2-NCOA2 and TEAD1-NCOA2 fusion proteins interact with CREBBP/EP300, recruiting them to promote TEAD target gene expression.

## EP300 is required for VGLL2-NCOA2 and TEAD1-NCOA2-induced tumorigenesis

To test whether VGLL2-NCOA2- and TEAD1-NCOA2-mediated sarcomagenesis is CREBBP/EP300 dependent, we utilized the C2C12 myoblast transformation models both in vitro and in vivo. Prior reports have demonstrated that VGLL2-NCOA2 can induce oncogenic transformation of C2C12 cells in cell culture and allograft mouse tumor models (*Watson et al., 2023*). However, it is not known whether VGLL2-NCOA2 or TEAD1-NCOA2 can induce TEAD-mediated transcription and whether EP300 is functionally relevant in these models.

We first demonstrated that VGLL2-NCOA2 binds to EP300 via its C-terminal NCOA2 part using co-immunoprecipitation assays (*Figure 5A*), highlighting the essential role of the NCOA2 part of the fusion proteins in recruiting EP300 to induce downstream transcription and tumorigenesis. We found that the expression of both VGLL2-NCOA2 and TEAD1-NCOA2 fusion proteins was able to robustly induce the transcription of the known Tead target genes, *Ccn1*, *Ccn2*, and *Ankrd1*, in C2C12 cells (*Figure 5B–D*). By utilizing a potent EP300 small-molecule inhibitor, A485, which inhibits CREBBP/EP300 activity both in vitro and in vivo (*Hogg et al., 2021*; *Lasko et al., 2017*), we showed that EP300 inhibition by A485 treatment strongly inhibited transcriptional upregulation of *Ccn1*, *Ccn2*, and *Ankrd1* induced by the fusion proteins (*Figure 5B–D*). In contrast, A485 was not able to strongly block YAP$^{5SA}$-induced Tead target gene expression in C2C12 cells (*Figure 5B–D*), suggesting that CREBBP/EP300 is required for VGLL2-NCOA2 and TEAD1-NCOA2-dependent gene expression but is largely dispensable for YAP/TAZ-induced target gene transcription. Furthermore, we demonstrated that VGLL2-NCOA2 and TEAD1-NCOA2 were also able to induce colony formation of C2C12 cells in soft agar (*Figure 5E*). More importantly, A485 treatment markedly reduced the numbers and sizes of

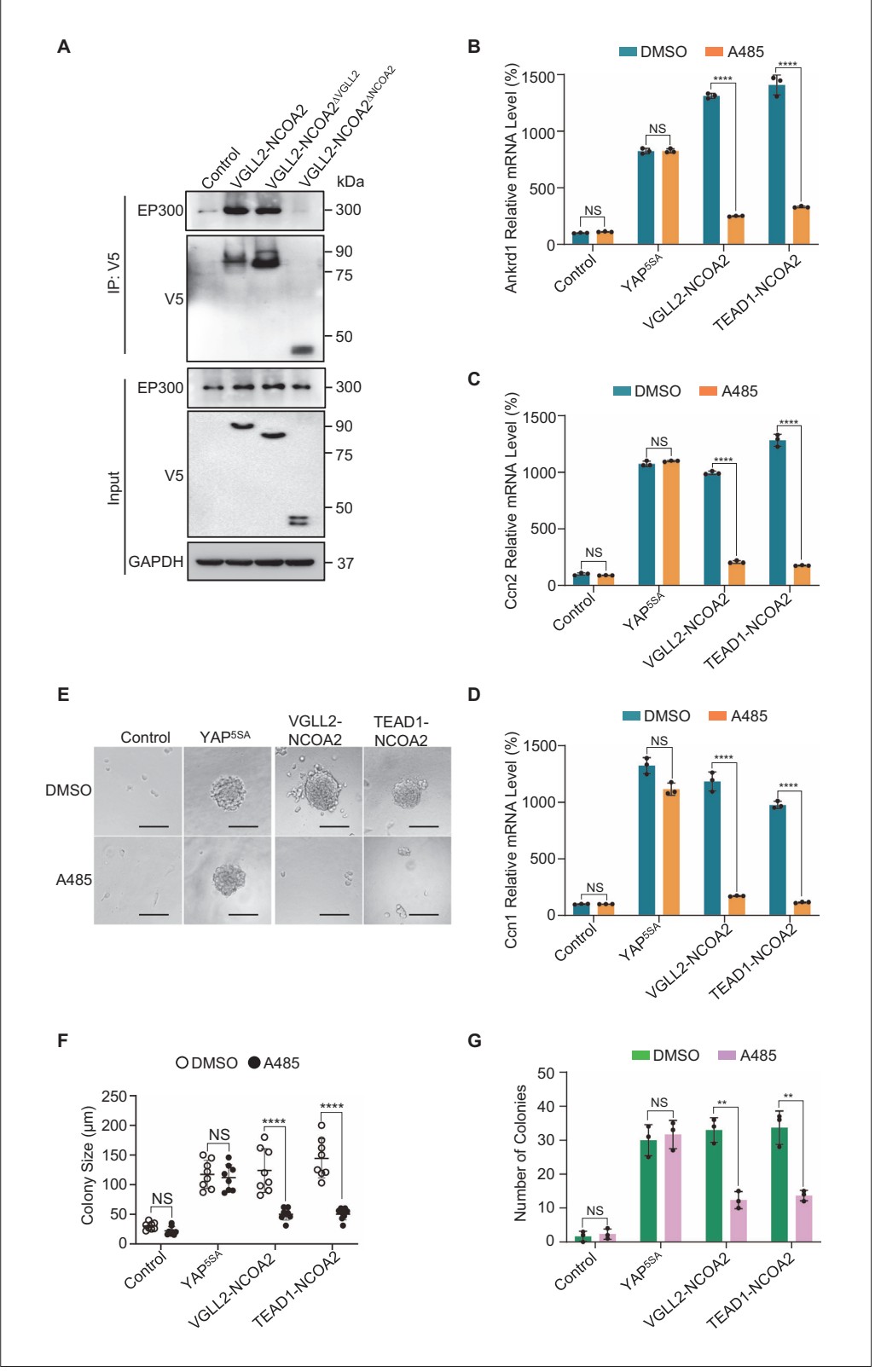

**Figure 5.** EP300 is required for VGLL2-NCOA2- and TEAD1-NCOA2-induced tumorigenesis in vitro. (**A**) Co-IP assays showing the NCOA2 fusion part of VGLL2-NCOA2 was essential for EP300 binding. VGLL2-NCOA2-V5, VGLL2-NCOA2$^{\Delta NCOA2}$-V5, and VGLL2-NCOA2$^{\Delta VGLL2}$-V5 were expressed in HEK293T cells and immunoprecipitated using an anti-V5 antibody. Endogenous EP300 proteins were detected by anti-EP300 antibody. (**B–D**) mRNA

*Figure 5 continued on next page*

*Figure 5 continued*

levels of *Ccn2*, *Ankrd1*, and *Ccn1* in C2C12 cells expressing YAP[5SA], VGLL2-NCOA2, or TEAD1-NCOA2 with or without treatment of A485 (5 µM). Data were expressed as mean ± SD. n=3; ****p<0.0001. NS, no significance. (**E**) Representative image of colony formation of C2C12 cells expressing YAP[5SA], VGLL2-NCOA2, or TEAD1-NCOA2 with or without treatment of A485 (5 µM) for 2 weeks. Scale bars, 100 µm. (**F, G**) Colony size (**F**) and number of colonies (**G**) formed by C2C12 cells expressing YAP[5SA], VGLL2-NCOA2, or TEAD1-NCOA2 with or without treatment of A485 (5 µM). Data were expressed as mean ± SD. **p<0.01; ****p<0.0001. NS, no significance.

The online version of this article includes the following source data for figure 5:

**Source data 1.** Original western blot membranes corresponding to *Figure 5A* indicating the relevant bands.

**Source data 2.** Original western blot membranes corresponding to *Figure 5A* indicating the relevant bands.

the soft agar colonies induced by TEAD1-NCOA2 and VGLL2-NCOA2 but not YAP[5SA] (*Figure 5F and G*), which is consistent with the differential effect of EP300 inhibition on downstream gene transcription induced by the fusion proteins or YAP (*Figure 5B–D*).

To examine the functional dependence of CREBBP/EP300 in tumorigenesis in vivo, we took advantage of the C2C12 allograft tumor models and demonstrated that both VGLL2-NCOA2 and TEAD1-NCOA2-expressing C2C12 cells were able to generate aggressive tumors when allografted into nude mice, leading to termination of tumor-bearing mice at around 30 days post-injection. These VGLL2-NCOA2- and TEAD1-NCOA2-expressing tumors were highly proliferative, measured by Ki67 staining, but exhibited heterogeneous expression of Desmin (*Figure 6A and B*). In addition, we found that EP300 inhibition by A485 significantly blocked the tumor growth in both VGLL2-NCOA2 and TEAD1-NCOA2 allograft tumor models (*Figure 6C and D*), and A485 treatment markedly decreased the proliferation of tumor cells (*Figure 6E and F*) and the transcription of the target genes, including *Ccn1*, *Ccn2*, and *Ankrd1* (*Figure 6G and H*), in both VGLL2-NCOA2 and TEAD1-NCOA2 tumor models. Together, these data, both in vitro and in vivo, suggest that the CREBBP/EP300 factors are essential for tumorigenesis induced by VGLL2-NCOA2 and TEAD1-NCOA2 fusion proteins.

## Discussion

Misregulation of YAP and TAZ, including protein upregulation, nuclear accumulation, and fusion translocation, has emerged as a major mechanism underlying Hippo signaling involvement in tumorigenesis (*Piccolo et al., 2023*; *Thompson, 2020*; *Franklin et al., 2023*; *Szulzewsky et al., 2021*; *Garcia et al., 2022*). In this work, our characterization of VGLL2-NCOA2 and TEAD1-NCOA2 fusion proteins generated by recurrent rearrangement in scRMS provides a distinct mechanism of oncogenic transformation that is related to the Hippo pathway but independent of the YAP/TAZ function.

Among the members of the VGLL family proteins, VGLL4 stands out as a *bona fide* transcriptional repressor for YAP/TAZ via its two TEAD-binding TDU domain. The other members of the family, VGLL1-3, contain a single TDU domain, and it is not fully understood whether and how they regulate YAP/TAZ-TEAD transcriptional output. Unlike YAP, TAZ, and VGLL2-NCOA2 fusion, VGLL2 itself does not appear to be a strong activator of TEAD-mediated transcription, even when overexpressed in cells. In both VGLL2-NCOA2 and TEAD1-NCOA2 fusions, the lack of NCOA2 N-terminal domains suggests that they are unlikely to be subject to downstream regulation by nuclear receptors, the canonical binding partners of the NCOA/SRC transcriptional regulators (*Yi et al., 2021*). Our previous work showed that the NCOA family proteins are associated with the YAP/TAZ-TEAD transcriptional machinery (*Liu et al., 2016*); however, the presence of YAP/TAZ in the complex appears to be dominant in directing TEAD-mediated transcriptional activation in the normal context. Clearly, the C-terminal NCOA2 TADs render robust transcriptional activity in both VGLL2-NCOA2 and TEAD1-NCOA2 fusions, underlying their oncogenic activity that bypasses the requirement for YAP/TAZ.

Our data showed that VGLL2-NCOA2 and TEAD1-NCOA2 fusions converge on the TEAD family transcription factors or TEAD-mediated genomic occupancy, highlighting the central role of TEADs in Hippo-related tumorigenesis. The small-molecule TEAD inhibitors are currently undergoing preclinical or clinical evaluation for cancer treatment (*Pobbati et al., 2023*). These inhibitors target TEAD autopalmitoylation, an enzymatic-like activity carried out by the C-terminal YAP/TAZ binding domain of the TEAD proteins (*Li et al., 2020*; *Pobbati et al., 2023*; *Chan et al., 2016*; *Holden et al., 2020*; *Kaneda et al., 2020*; *Tang et al., 2021*; *Hu et al., 2022*). The lack of YAP/TAZ binding domain in TEAD1 fusion

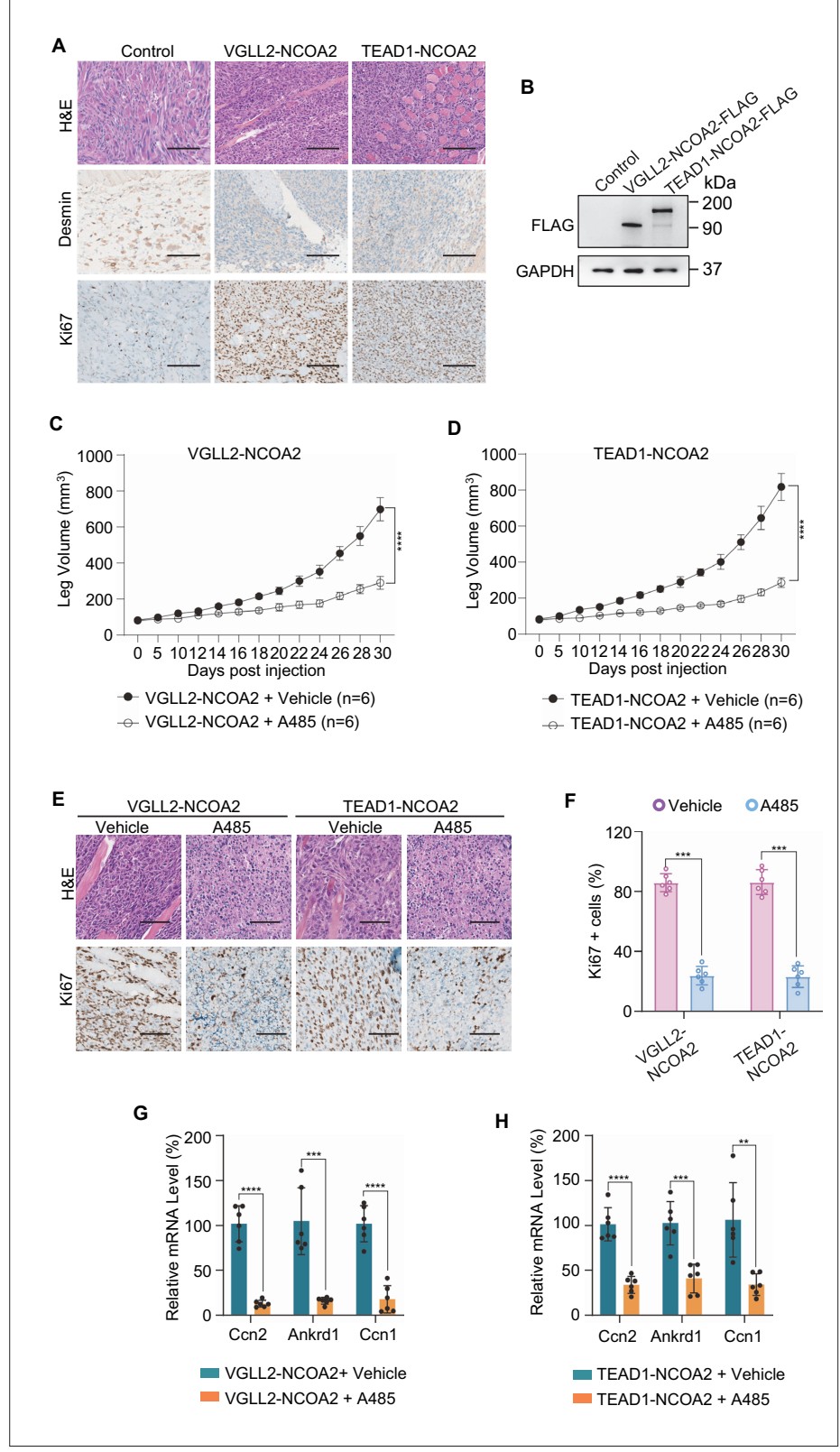

**Figure 6.** EP300 is essential for VGLL2-NCOA2- and TEAD1-NCOA2-induced tumorigenesis in vivo.
(**A**) Representative H&E and IHC staining of Desmin and Ki67 in C2C12-control allograft, C2C12-VGLL2-NCOA2 tumor allograft, and C2C12-TEAD1-NCOA2 tumor allograft. Scale bars, 200 μm. (**B**) Immunoblot analysis of VGLL2-NCOA2-FLAG and TEAD1-NCOA2-FLAG expression in C2C12 cells, detected by an anti-FLAG antibody. (**C,**

*Figure 6 continued on next page*

*Figure 6 continued*

**D**) Allograft leg volume of C2C12-VGLL2-NCOA2 (**C**) and C2C12-TEAD1-NCOA2 (**D**) after intramuscular injection into the leg of Nude mice with or without intraperitoneal injection of A485 (100 mg/kg). The error bars represent the mean leg volume ± SEM. n=6. ****p<0.0001. (**E**) Representative H&E and IHC staining of Ki67 in C2C12-VGLL2-NCOA2 and C2C12-TEAD1-NCOA2 tumor allografts with or without A485 (100 mg/kg) treatment. Scale bars, 200 µm. (**F**) Percentage of Ki67-positive cells in (**E**). Data were expressed as mean ± SD. n=6; ***p<0.001. (**G, H**) mRNA levels of *Ccn2*, *Ankrd1*, and *Ccn1* in C2C12-VGLL2-NCOA2 and C2C12-TEAD1-NCOA2 tumor allografts with or without A485 (100 mg/kg) treatment. Data were expressed as mean ± SD. **p<0.01; ***p<0.001; ****p<0.0001.

The online version of this article includes the following source data for figure 6:

**Source data 1.** Original western blot membranes corresponding to *Figure 6B* indicating the relevant bands.

**Source data 2.** Original western blot membranes corresponding to *Figure 6B* indicating the relevant bands.

results in the inability of the TEAD palmitoylation inhibitor to suppress gene transcription, suggesting the potential limitation of TEAD palmitoylation inhibitors in tumors carrying TEAD rearrangements. In addition, current TEAD small-molecule inhibitors are generally thought to allosterically affect the binding between YAP/TAZ and TEADs. How these inhibitors may affect TEAD interaction with VGLL or VGLL fusion proteins requires further investigation.

Our identification of CREBBP/EP300 as specific epigenetic regulators for VGLL2-NCOA2 and TEAD1-NCOA2 proteins provides a molecular mechanism underlying their YAP/TAZ-independent regulation of transcription and tumorigenesis. In another recurrent *VGLL2* rearrangement identified in scRMS, *CITED2*, the 3′ partner in the VGLL2-CITED2 fusion, is a known transcriptional regulator associated with CREBBP/EP300 (*Alaggio et al., 2016*; *Bhattacharya et al., 1999*; *Bamforth et al., 2001*). It is possible that the VGLL2-CITED2 fusion also mediates YAP/TAZ-independent TEAD-dependent transcriptional activation by engaging CREBBP/EP300. Our data using small-molecular CREBBP/EP300 inhibitors both in vitro and in vivo further supports its critical role in VGLL and TEAD fusion protein-mediated oncogenic transformation. This finding may also provide a potential therapeutic strategy for scRMS characterized by these fusion proteins, as well as a range of other tumors carrying NCOA1/2/3 rearrangements, including uterine sarcoma (*Niu et al., 2023*), mesenchymal chondrosarcoma (*Wang et al., 2012*; *Tanaka et al., 2023*), soft tissue angiofibroma (*Jin et al., 2012*; *Yamashita et al., 2023*), ovarian sex cord tumors (*Goebel et al., 2020*; *Lu et al., 2023*), leukemia (*Strehl et al., 2008*), colon cancer (*Yu et al., 2016*), and ependymoma (*Tauziède-Espariat et al., 2021*; *Tomomasa et al., 2021*).

# Materials and methods

## Key resources table

| Reagent type (species) or resource | Designation | Source or reference | Identifiers | Additional information |
|---|---|---|---|---|
| Antibody | Anti-Flag (mouse monoclonal) | Invitrogen | MA1-91878 | CUT&RUN (0.5 µg) |
| Antibody | Anti-GAPDH (rabbit monoclonal) | Cell Signaling Technology | 2118 | WB (1:10,000) |
| Antibody | Anti-YAP/TAZ (rabbit monoclonal) | Cell Signaling Technology | 8418 | WB (1:1000) |
| Antibody | Anti-panTEAD (rabbit monoclonal) | Cell Signaling Technology | 13295 | WB (1:1000) |
| Antibody | Anti-V5 (rabbit monoclonal) | Cell Signaling Technology | 13202 | WB (1:1000) |
| Antibody | Anti-EP300 (rabbit monoclonal) | Cell Signaling Technology | 86377 | WB (1:1000); CUT&RUN (0.5 µg) |
| Antibody | Anti-Flag (rabbit monoclonal) | Cell Signaling Technology | 2368 | WB (1:1000); IP (1:50) |

*Continued on next page*

*Continued*

| Reagent type (species) or resource | Designation | Source or reference | Identifiers | Additional information |
|---|---|---|---|---|
| Antibody | Anti-Flag (mouse monoclonal) | Cell Signaling Technology | F9291 | WB (1:1000); IP (1:50) |
| Antibody | Anti-HA (rabbit monoclonal) | Cell Signaling Technology | 3724 | WB (1:1000); IP (1:50) |
| Antibody | Anti-HA (mouse monoclonal) | Cell Signaling Technology | 2367 | WB (1:1000); IP (1:50) |
| Antibody | HRP-conjugated secondary antibodies | Jackson Laboratories | | WB (1:2000) |
| Antibody | Anti-Ki67 (rabbit monoclonal) | Cell Signaling Technology | 12202 | IHC (1:500) |
| Antibody | Anti-Desmin (rabbit monoclonal) | Cell Signaling Technology | 5332 | IHC (1:100) |
| Strain, strain background (NU/J nude mice) | NU/J nude mice | The Jackson Laboratory | 002019 | |
| Cell line (*Homo sapiens*) | HEK293T | ATCC | CRL-3216 | |
| Cell line (mouse-*sapiens*) | C2C12 | ATCC | CRL-1772 | |
| Chemical compound, drug | A485 | SelleckChem | S8740 | |
| Chemical compound, drug | Streptavidin Sepharose beads | GE Healthcare | 17511301 | |
| Chemical compound, drug | SignalStain Antibody Diluent | Cell Signaling Technology | 8112 | |
| Chemical compound, drug | Lipofectamine 2000 | Invitrogen | 11668019 | |
| Chemical compound, drug | CP1 | MCE | HY-139330 | |
| Commercial assay or kit | CUTANA ChIC/CUT&RUN Kit | EpiCypher | 14-1048 | |
| Commercial assay or kit | CUT&RUN Library Prep Kit | EpiCypher | 14-1001 | |
| Commercial assay or kit | RNeasy Mini Kit | QIAGEN | 74104 | |
| Commercial assay or kit | Vectastain Elite ABC kit | Vector Laboratories | PK-6105 | |
| Commercial assay or kit | In-Fusion HD Cloning | Clontech | Clontech: 639647 | |
| commercial assay or kit | Dual-luciferase reporter kit | Promega | E1910 | |
| Commercial assay or kit | iTaq Universal One-Step RT-qPCR Kit | Bio-Rad | 1725150 | |
| Commercial assay or kit | BeyoClick EdU Cell Proliferation Kit with Alexa Fluor 488 | Beyotime | C0071 | |
| Commercial assay or kit | iTaq Universal One-Step RT-qPCR Kit | Bio-Rad | 1725150 | |
| Recombinant DNA reagent | *GAPDH*-F' | This paper | PCR primers | GGAGCGAGATCCCTCCAAAAT |
| Recombinant DNA reagent | *GAPDH*-R' | This paper | PCR primers | GGCTGTTGTCATACTTCTCATGG |
| Recombinant DNA reagent | *CCN2*-F' | This paper | PCR primers | CAGCATGGACGTTCGTCTG |
| Recombinant DNA reagent | *CCN2*-R' | This paper | PCR primers | AACCACGGTTTGGTCCTTGG |

*Continued*

| Reagent type (species) or resource | Designation | Source or reference | Identifiers | Additional information |
|---|---|---|---|---|
| Recombinant DNA reagent | *ANKRD1*-F' | This paper | PCR primers | GCCTACGTTTCTGAAGGCTG |
| Recombinant DNA reagent | *ANKRD1*-R' | This paper | PCR primers | GTGGATTCAAGCATATCACGGAA |
| Recombinant DNA reagent | *CCN1*-F' | This paper | PCR primers | CAGGACTGTGAAGATGCGGT |
| Recombinant DNA reagent | *CCN1*-R' | This paper | PCR primers | GCCTGTAGAAGGGAAACGCT |
| Recombinant DNA reagent | *Actb*-F' | This paper | PCR primers | GTGACGTTGACATCCGTAAAGA |
| Recombinant DNA reagent | *Actb*-R' | This paper | PCR primers | GCCGGACTCATCGTACTCC |
| Recombinant DNA reagent | *Ccn2*-F' | This paper | PCR primers | GACCCAACTATGATGCGAGCC |
| Recombinant DNA reagent | *Ccn2*-R' | This paper | PCR primers | CCCATCCCACAGGTCTTAGAAC |
| Recombinant DNA reagent | *Ankrd1*-F' | This paper | PCR primers | GGATGTGCCGAGGTTTCTGAA |
| Recombinant DNA reagent | *Ankrd1*-R' | This paper | PCR primers | GTCCGTTTATACTCATCGCAGAC |
| Recombinant DNA reagent | *Ccn1*-F' | This paper | PCR primers | TAAGGTCTGCGCTAAACAACTC |
| Recombinant DNA reagent | *Ccn1*-R' | This paper | PCR primers | CAGATCCCTTTCAGAGCGGT |

## Cell lines, transfection, lentiviral infections and luciferase reporter assays

HEK293T (cat. CRL-3216, ATCC, Manassas, VA, USA) and C2C12 (cat. CRL-1772, ATCC) cells were cultured in DMEM supplemented with 10% fetal bovine serum (FBS) and 1% penicillin/streptomycin. Transfection in HEK293T cells was performed using Lipofectamine 2000 (cat. 11668019, Invitrogen, Waltham, MA, USA). For luciferase reporter assays, HEK293T cells were transfected with the luciferase reporter construct TBS-Luc (8XGTIIC-Luc, cat. 34615, Addgene, 8xGTIIC-luc was a gift from Stefano Piccolo), and the expression vectors for YAP$^{5SA}$, TAZ$^{4SA}$, VGLL2, NCOA2, TEAD1, TEAD-ENR, VGLL2-NCOA2, VGLL2-NCOA2$^{\Delta VGLL2}$, VGLL2-NCOA2$^{\Delta NCOA2}$, and TEAD1-NCOA2 (generated by and purchased from GenScript, Piscataway, NJ, USA; and GentleGen, Suzhou, Jiangsu Province, China) and pCMV-Renilla luciferase. Luciferase activities were conducted 24 hours after transfection in the cells treated with or without the TEAD inhibitor CP1 (5 μM for 24 hours) using the dual-luciferase reporter kit (cat. E1910, Promega, Madison, WI, USA). Assays were conducted in triplicates and quantified using PerkinElmer EnVision plate reader. For lentiviral infection, pLKO or pLX-based constructs expressing VGLL2-NCOA2, TEAD1-NCOA2, or shRNAs against YAP (5'-CCGGAAGCTTTGAGTT CTGACATCCCTCGAGGGATGTCAGAACTCAAAGCTTTTTTTC-3', cat. 27368, Addgene, pLKO1-shYAP1 was a gift from Kunliang Guan) or TAZ (TRCN0000019470, purchased from Sigma, St. Louis, MO, USA) were transfected along with the packaging plasmids into growing HEK293T cells. Viral supernatants were collected 48 hours after transfection, and target cells were infected in the presence of polybrene and underwent selection with puromycin for 4–5 days. All cell lines used in this study were authenticated by short tandem repeat (STR) profiling and confirmed to be free of Mycoplasma contamination using a Mycoplasma detection assay.

## Protein immunoprecipitation and western blot

Cultured HEK293T cells were lysed in lysis buffer (50 mM Tris–HCl, pH 7.4, 150 mM NaCl, 0.5 mM EDTA, 1% Triton X-100, phosphatase inhibitor cocktail, cOmplete EDTA-free protease inhibitors

cocktail) for 30 minutes at 4°C. The supernatants of the extracts were then used for the immunoprecipitation and western blot following the protocols described previously (*Cotton et al., 2017*). The primary antibodies used in these assays were: GAPDH (cat. 2118, 1:5000, Cell Signaling Technology, Danvers, MA, USA), YAP/TAZ (cat. 8418, 1:1000, Cell Signaling Technology), panTEAD (cat. 13295, 1:1000, Cell Signaling Technology), V5-tag (cat. 13202, 1:1000, Cell Signaling Technology), EP300 (cat. 86377, 1:1000, Cell Signaling Technology), Flag-tag (cat. 2368, 1:1000, Cell Signaling Technology and cat. F9291, 1:1000, Sigma), and HA-tag (cat. 3724 and cat. 2367, 1:1000, Cell Signaling Technology). HRP-conjugated secondary antibodies used for detection were obtained from Jackson Laboratories (Sacramento, CA, USA).

## Quantitative RT-PCR (qPCR)

Total RNA was extracted using the RNeasy Mini Kit (cat. 74104, QIAGEN, Germantown, MD, USA). Gene expression was quantified using the iTaq Universal One-Step RT-qPCR Kit (cat. 1725150, Bio-Rad, Hercules, CA, USA) in Applied Biosystems and normalized to *GAPDH* or *Actb*. The primers used in this study are provided in the Key Resources Table. qPCR experiments were performed in triplicate, with average cycle threshold (Ct) values from three independent reactions used for analysis.

## RNA sequencing (RNA-seq)

Total RNA was isolated with the RNeasy Mini Kit (cat. 74104, QIAGEN). The integrity of the isolated RNA and RNA-seq libraries was analyzed by Novogene. All libraries had at least 50 million reads sequenced (150 bp paired-end). The correlation between gene expression changes was performed using Pearson's correlation analysis. The p-values of gene changes in control groups compared with the experimental group were determined by Student's *t*-test. Plots of correlation between fold change were generated using the ggplot2 package in R. Principal component analysis was determined and plotted using the M3C package in R. Gene Set Enrichment Analysis was performed using GSEA software. We performed RNA-seq in triplicate for each condition, and each replicate was independently processed and analyzed.

## Cleavage under targets and release using nuclease (CUT&RUN)

CUT&RUN was performed using the EpiCypher kit (cat. 14-1048, EpiCypher, Durham, NC, USA). Approximately 0.5 million cells were used for each reaction. In brief, HEK293T cells expressing YAP$^{5SA}$ or VGLL2-NCOA2, or TEAD1-NCOA2 were attached to preactivated ConA beads for 10 minutes at room temperature. Then, 0.5 µg of antibodies against Flag (cat. MA1-91878, Invitrogen) or EP300 (cat. 86377, Cell Signaling Technology) as well as control IgG were added to the reactions and incubated overnight at 4°C. The cells were incubated with pAG-MNase for 10 minutes at room temperature prior to activation by calcium chloride. Subsequently, the cells were incubated at 4°C for 2 hours prior to the addition of stop buffer. DNA was purified for library construction using a CUT&RUN Library Prep Kit (cat. 14-1001, EpiCypher). Illumina HiSeq sequencing of approximately 10 million paired-end 150 bp reads was performed at Novogene. MACS2 software (v2.2.7.1) was used for peak calling. Deeptools software (v3.5.1) was used to plot heatmaps of the distribution of reads around TSS. Homer software (v4.11) was used to identify motifs. MANorm2 software (v1.2.0) was used to identify differentially enriched regions between two samples and/or two groups of samples. The scatter plot of the comparison of the two groups of samples is plotted according to the standardized expression mean. The enriched peaks were visualized in IGV (v2.4.10) software. GO and KEGG Pathway analysis were performed based on the promoter enriched peaks associated genes by the ClusterProfiler R packages (v3.18.1). CUT&RUN assays were performed in duplicate for each condition and each replicate was independently processed and analyzed.

## Assay for transposase-accessible chromatin (ATAC-seq)

ATAC-seq library construction was performed as previously reported (*Buenrostro et al., 2013*; *Bajic et al., 2018*). Briefly, nuclei were extracted from HEK293T cells expressing GFP or VGLL2-NCOA2, and the nuclei pellet was resuspended in the Tn5 transposase reaction mix. The transposition reaction was incubated at 37°C for 30 minutes. Equimolar Adapter 1 and Adapter 2 were added after transposition, and PCR was then performed to amplify the library. After PCR amplification, libraries were purified with AMPure beads. The library was checked with Qubit and real-time PCR for quantification, and

bioanalyzer for size distribution detection. Quantified libraries were pooled and sequenced on Illumina platforms, according to the effective library concentration and data amount required. ATAC-seq bioinformatics data analysis was performed following Harvard FAS Informatics ATAC-seq Guidelines. Briefly, Genrich software (v1.30.3) was used for peak calling and blacklist regions were removed. Deeptools software (v3.5.1) was used to plot heatmaps of the distribution of reads around TSS. ChIP-seeker R package (v1.30.3) was used to annotate the enriched peaks. Homer software (v4.11) was used to identify motifs. MANorm2 R package was used to perform differentially enriched region analysis between two groups of samples. The scatter plot of the comparison of the two groups of samples is plotted according to the standardized expression mean. GO and KEGG pathway enrichment analysis were performed with ClusterProfiler R package (v3.18.1) based on the enriched peak and differentially enriched region-associated genes. ATAC-seq was performed in duplicate for each condition, and each replicate was independently processed and analyzed.

## BioID proximity ligation and mass spectrometry analysis

The expression vectors *pGIPZ-nlsGFP*, *pGIPZ-nlsYAP5SA*, *pGIPZ-nlsTAZ4SA*, *pGIPZ-VGLL2-NCOA2*, and *pGIPZ-TEAD1-NCOA2* generated by PCR sub-cloning into the *pGIPZ* vector were transiently transfected into HEK293T cells. The biotin treatment and BioID pull-down procedures were performed as described previously (*Roux et al., 2012*). Briefly, 24 hours after transfection, the cells at ~80% confluence were treated with biotin at 50 µM for 24 hours. After washing with cold PBS, the cells were lysed in lysis buffer (50 mM Tris–HCl, pH 7.5, 150 mM NaCl, 1% NP-40, 1 mM EDTA, 1 mM EGTA, 0.1% SDS, 0.5% sodium deoxycholate, 250 U of benzonase, and protease inhibitor), and the cell lysate was incubated at 4°C for 1 hour and sonicated on ice three times. The supernatant was then mixed with pre-washed streptavidin Sepharose beads (cat. 17511301, GE Healthcare, Bronx, NY, USA) and incubated at 4°C with rotation for 3 hours. After affinity purification, the agarose beads were washed with lysis buffer followed by two washes with TAP buffer (50 mM HEPES-KOH, pH 8.0, 100 mM KCl, 10% glycerol, 2 mM EDTA, and 0.1% NP-40) and three washes with ABC buffer (50 mM ammonium bicarbonate, pH 8.0). Protein samples were then eluted by boiling for 5 minutes at 95°C in SDS-containing buffer and collected from SDS-PAGE gels before subjecting to mass spectrometry analysis. MS data were searched against a protein database (UniProt KB) using the Mascot search engine program (Matrix Science, London, UK) for protein identification. MS data were validated using the Scaffold 4 program (Proteome Software Inc, Portland OR, USA).

## Soft agar colony formation and EdU-based proliferation assays

The soft agar colony formation protocol was described in *Borowicz et al., 2014*. Briefly, in a well of a 6-well plate, a bottom layer of 1% agar in growth media (DMEM + 10% FBS) was plated and solidified, followed by a top layer of 0.6% agar + growth media containing 5000 C2C12 cells with or without expressing VGLL2-NCOA2 or TEAD1-NCOA2. A485 (cat. S8740, SelleckChem, Houston, TX, USA) was dissolved in DMSO to prepare 5 mM stock solution. 1 ml of growth media with or without A485 (5 µM working concentration) was added to the top of each well and replenished every 2–3 days. Colony growth was monitored under a microscope for 2 weeks. Colonies from each group were imaged using a Zeiss Axio Photo Observer microscope. Eight colonies from each group were randomly selected for size measurement, and the diameter of each colony was measured using ZEN 3.2 blue edition software. Colony numbers were counted from three random fields of view and representative images were shown.

Cell proliferation was assessed using the BeyoClick EdU Cell Proliferation Kit with Alexa Fluor 488 (Beyotime, C0071) according to the manufacturer's protocol. Briefly, cells were incubated with 10 µM EdU for 120 minutes, followed by digestion with 0.25% trypsin-EDTA. The cells were then fixed in 4% paraformaldehyde (PFA) and permeabilized with PBS containing 0.3% Triton X-100 at room temperature for 15 minutes. After three washes with PBS containing 3% BSA, a click reaction solution was added and incubated at room temperature for 30 minutes, followed by Hoechst 33342 counterstaining. EdU detection assays were performed in triplicate, and representative images were shown.

## Mouse allograft experiments and tumor processing

All animal protocols and procedures were approved by the institutional animal care and use committees at the Shanghai Chest Hospital and the University of Massachusetts Chan Medical School. Nude

mice were purchased from The Jackson Laboratory (male, 3–4 weeks, NU/J, cat. 002019, Shanghai, China). Nude mice were randomly divided into groups. C2C12-control, C2C12-VGLL2-NCOA2, and C2C12-TEAD1-NCOA2 cell lines were utilized in allograft experiments adapted from *Watson et al., 2023*. Five million cells were resuspended in 100 µl of sterile PBS and injected intramuscularly into the leg of nude mice under anesthesia. Each experimental group had six mice, and mouse legs were assessed 5 days after initial injection. Based on the weight of each mouse, an appropriate volume of the A485 stock solution (5 mM in DMSO) was diluted in 500 µl saline and then administered daily via intraperitoneal injection at a dose of 100 mg/kg (*Lasko et al., 2017*). The height and width of the injected legs were measured using calipers to calculate the volume until 30 days post-injection (initial leg volume was 50–80 mm$^3$). Tumor tissue was snap-frozen in liquid nitrogen, and RNA was isolated with Trizol for qPCR analysis. For immunohistochemistry (IHC), tumors were fixed in a 4% PFA solution and mounted in paraffin blocks for microtomy.

## Immunohistochemistry

Sections were deparaffinized and rehydrated before undergoing heat-induced antigen retrieval in 10 mM sodium citrate buffer (pH 6.0, Solarbio, cat. C1013) for 30 minutes. Slides were blocked for endogenous peroxidase for 20 minutes, then blocked for 1 hour in 5% BSA, 1% goat serum, 0.1% Tween-20 buffer in PBS, and incubated overnight at 4°C with the primary antibody diluted in blocking buffer or SignalStain Antibody Diluent (cat. 8112, Cell Signaling Technology). Slides were incubated with biotinylated secondary antibodies for 1 hour at room temperature, and the signal was detected using the Vectastain Elite ABC kit (cat. PK-6105, Vector Laboratories, Newark, CA, USA). Hematoxylin was used for counterstaining in IHC. Antibodies and dilutions used in the studies: Ki67 (cat. 12202, 1:500, Cell Signaling Technology) and Desmin (cat. 5332, 1:100, Cell Signaling Technology).

## Statistics and reproducibility

Statistical analyses were performed using GraphPad Prism 8 software or R. The repetition of the experiment is shown in the figures and corresponding figure legends. The results were expressed as the mean ± SEM. Statistical significance was determined using a one-way ANOVA for most datasets. Student's *t*-test was used when comparing two groups.

## Acknowledgements

This work was supported by grants from the National Institutes of Health (R01DK127207 and R01DK127180 to JM, R01CA238270 to XW and JM) and the National Natural Science Foundation of China (82173015 to JW). We also thank the members of the Wang lab and Mao lab for their helpful discussions.

## Additional information

### Funding

| Funder | Grant reference number | Author |
| --- | --- | --- |
| National Institutes of Health | R01DK127207 | Junhao Mao |
| National Institutes of Health | R01DK127180 | Junhao Mao |
| National Institutes of Health | R01CA238270 | Xu Wu |
| National Natural Science Foundation of China | 82173015 | Jiayi Wang |

The funders had no role in study design, data collection and interpretation, or the decision to submit the work for publication.

## Author contributions
Susu Guo, Conceptualization, Data curation, Formal analysis, Investigation, Methodology, Validation, Writing – original draft, Writing – review and editing; Xiaodi Hu, Conceptualization, Data curation, Validation, Visualization, Methodology; Jennifer L Cotton, Lifang Ma, Qi Li, Jiangtao Cui, Yongjie Wang, Ritesh P Thakare, Zhipeng Tao, Validation; Y Tony Ip, Funding acquisition; Xu Wu, Data curation, Funding acquisition; Jiayi Wang, Junhao Mao, Investigation, Data curation, Supervision, Funding acquisition, Writing – original draft, Project administration, Writing – review and editing

## Author ORCIDs
Susu Guo (ID) http://orcid.org/0000-0002-7481-6884
Jennifer L Cotton (ID) https://orcid.org/0000-0003-0106-1801
Y Tony Ip (ID) https://orcid.org/0000-0003-4370-7906
Jiayi Wang (ID) http://orcid.org/0000-0003-1688-2864
Junhao Mao (ID) https://orcid.org/0000-0003-1980-1177

## Ethics
All animal protocols and procedures were approved by the animal ethics committee at the Shanghai Chest Hospital (reference numbers: KS(Y)24388).

Reviewer #1 (Public review): https://doi.org/10.7554/eLife.98386.3.sa1
Reviewer #2 (Public review): https://doi.org/10.7554/eLife.98386.3.sa2
Author response https://doi.org/10.7554/eLife.98386.3.sa3

---

# Additional files

## Supplementary files
Supplementary file 1. RNA-seq analysis showing the genes whose expression was significantly changed in HEK293T cells expressing VGLL2-NCOA2, p-value <0.05, Log2 Fold Change >1 or Log2 Fold Change < –1.

Supplementary file 2. RNA-seq analysis showing the genes whose expression was significantly changed in HEK293T cells expressing TEAD1-NCOA2, p-value <0.05, Log2 Fold Change >1 or Log2 Fold Change < –1.

Supplementary file 3. RNA-seq analysis showing the genes whose expression was significantly changed in HEK293T cells expressing YAP5SA, p-value <0.05, Log2 Fold Change >1 or Log2 Fold Change < –1.

Supplementary file 4. List of specific BioID hits obtained with BirA*-YAP[5SA], BirA*-TAZ[4SA], BirA*-VGLL2-NCOA2, and BirA*-TEAD1-NCOA2.

MDAR checklist

## Data availability
Sequencing data have been deposited in Gene Expression Omnibus (GEO) under accession codes GSE260781, GSE260782, and GSE260783.

The following datasets were generated:

| Author(s) | Year | Dataset title | Dataset URL | Database and Identifier |
|---|---|---|---|---|
| Mao J, Guo S, Hu X ML | 2024 | Effect of VGLL2-NCOA2, YAP5SA and GFP overexpression on chromatin accessibility profiling in HEK293T cells (ATAC-Seq) | https://www.ncbi.nlm.nih.gov/geo/query/acc.cgi?acc=GSE260781 | NCBI Gene Expression Omnibus, GSE260781 |

*Continued on next page*

*Continued*

| Author(s) | Year | Dataset title | Dataset URL | Database and Identifier |
|---|---|---|---|---|
| Mao J, Guo S, Hu X ML | 2024 | VGLL2-NCOA2 and TEAD1-NCOA2 fusions recruited p300 to drive TEAD-dependent transcription (CUT&RUN) | https://www.ncbi. nlm.nih.gov/geo/ query/acc.cgi?acc= GSE260782 | NCBI Gene Expression Omnibus, GSE260782 |
| Mao J, Guo S, Hu X ML | 2024 | Effect of VGLL2-NCOA2, TEAD1-NCOA2 and YAP5SA overexpression on mRNA expression in HEK293T cells (RNA-Seq) | https://www.ncbi. nlm.nih.gov/geo/ query/acc.cgi?acc= GSE260783 | NCBI Gene Expression Omnibus, GSE260783 |

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
