## [Editor Report · eLife Assessment]

This is a **valuable** study describing how rhabdomyosarcoma fusion-oncogenes, VGLL2-NCOA2 and TEAD1-NCOA2, function at the genomic, transcriptional, and proteomic levels in multiple systems. The experimental data is **convincing**, supporting a model in which these fusion-oncogenes leverage TEAD transcriptional signatures independent of YAP/TAZ. This work offers new mechanistic insights into oncogenic gene fusion events and reveals potential therapeutic strategies for the treatment of rhabdomyosarcomas.

---

## [Referee Report · Reviewer #1 (Public review)]

Guo, Hue et al., is focused on understanding the epigenetic activity and functional dependencies for two different fusions found in spindle cell rhabdomyosarcoma, VGLL2::NCOA2 and TEAD1::NCOA2. They use a variety of models and methods; specifically, ectopic expression of the fusions in human 293T cells to perform RNAseq (both fusions), CUT&RUN (VGLL2::NCOA2) and BioID mass spec (both fusions). These data identify that the VGLL2::NCOA2 fusion has peaks that are enriched for TEAD motifs. Further, CREBBP/EP300 CUT&RUN support an enrichment of binding sites and three TEAD targets in VGLL2::NCOA2 and TEAD1::NCOA2 expressing cells. They also functionally evaluate genetic and chemical dependencies (TEAD inhibition), and found this was only effective for the VGLL2::NCOA2 fusion, and not for TEAD1::NCOA2. Using complementary biochemical approaches, they suggest (with other supporting data) the fusions regulate TEAD transcriptional outputs via a YAP/TAZ independent mechanism. Further, they expand into a C2C12 myoblast model and show that TEAD1::NCOA2 is transforming in colony formation assays and in mouse allograft. These strategies for TEAD1-NCOA2 are consistent with previous published strategies using VGLL2::NCOA2. Importantly, they show that a CREBBP/EP300 (a binding partner found in their BioID mass spec) small molecule inhibitor suppresses tumor formation using this mouse allograft model, and that the tumors are less proliferative, and have a reduction in transcriptional of three TEAD target genes. They complement in vivo data with biochemical approaches, and suggest this interface with EP300 (for VGLL2::NCOA2) is through the NCOA2 fusion partner, as Co-IP in HEK293T with a mutant fusion that does not contain NCOA2 loses the association with endogenous EP300. The data is interesting and suggests new biology for these fusion-oncogenes. However, the choice of 293T may limit the broad applicability of the findings. Strikingly, in 293T there was more transcriptional overlap with the VGLL2-NCOA2 fusion with the YAP5SA mutant than with TEAD1-NCOA2. Further, there is an additional opportunity to directly compare transcriptional profiles in 293T to the human disease and in the mouse allograft system to directly compare and discuss VGLL2-NCOA2 and TEAD1-NCOA2 histological differences or how A485 treatment may change the histology. Overall, the breadth of methods used in this study, and comparison of the two fusion-oncogene's biology is of interest to the fusion-oncogene, pediatric sarcoma, and epigenetic therapeutic targeting fields.

---

## [Referee Report · Reviewer #2 (Public review)]

In the manuscript entitled "VGLL2 and TEAD1 fusion proteins drive YAP/TAZ-independent transcription and tumorigenesis by engaging p300", Gu et al. investigated two Hippo pathway-related gene fusion events (i.e., VGLL2-NCOA2, TEAD1-NCOA2) in spindle cell rhabdomyosarcoma (scRMS). They demonstrate that these fusion proteins activate Hippo downstream gene transcription independently of YAP/TAZ. Using BioID-based mass spectrometry analysis, the authors identify histone acetyltransferase CREBBP/EP300 as a specific binding protein for VGLL2-NCOA2 and TEAD1-NCOA2 fusion proteins. Pharmacologically targeting EP300 inhibits the fusion proteins-induced Hippo downstream gene transcription and tumorigenesis.

Overall, this work provides novel mechanistic insights into scRMS-associated gene fusions in tumorigenesis and reveals potential therapeutic targets for cancer treatment. The manuscript is well-written and easy to follow. Below are a few comments based on the revised study.

(1) While the study majorly focuses on Hippo downstream gene transcription, a significant portion of genes regulated by the VGLL2-NCOA2 and TEAD1-NCOA2 fusion proteins are non-Hippo downstream genes (Fig. 3). Further characterization of how both Hippo and non-Hippo downstream genes contribute to fusion proteins-induced oncogenesis would enhance our understanding of scRMS etiology.

(2) A potential limitation of this study is the reliance on overexpression approaches to investigate VGLL2-NCOA2 and TEAD1-NCOA2 fusion genes, which may not fully reflect pathological conditions in scRMS patients. Despite this, the significant study offers valuable mechanistic insights into fusion genes-induced scRMS and provides molecular foundation for developing targeted therapies.

---

## [Author Response]

The following is the authors’ response to the original reviews

**Reviewer #1 (Public Review):**
(1) The rationale for performing genomics, transcriptional, and proteomics work in 293T cells is not discussed. Further, there are no functional readouts mentioned in the 293T cells with expression of the fusion-oncogenes. Did these cells have any phenotypes associated with fusion-oncogene expression (proliferation differences, morphological changes, colony formation capacity)? Further, how similar are the gene expression signatures from RNA-seq to rhabdomyosarcoma? This would help the reader interpret how similar these cell models are to human disease.

We appreciate the reviewer’s comments and understand the limitation of HEK293T cell culture. HEK293T cells were used as a surrogate system that enabled us to systemically examine and compare the transcriptional activation mechanisms between VGLL2-NCOA2/TEAD1-NCOA2 and YAP/TAZ. HEK293T cells have previously been used as a model system to study the signaling and transcriptional mechanisms of the Hippo/YAP pathway (1,2). Our data also showed that the ectopic expression of VGLL2-NCOA2 and TEAD1-NCOA2 in HEK293 cells can promote proliferation (Figure 1-figure supplement 1B), consistent with their potential oncogenic function.

(2) TEAD1::NCOA2 fusion-oncogene model was not credentialed past H&E, and expression of Desmin. Is the transcriptional signature in C2C12 or 293T similar to a rhabdomyosarcoma gene signature?

We understand the reviewer’s concern. VGLL2-NCOA2 in vivo tumorigenesis model generated by C2C12 cell orthotopic transplantation has recently been reported, and it exhibits similar characteristics with zebrafish transgenic tumors as well as human scRMS samples that carry the VGLL2-NCOA2 fusion (3). Due to the similar transcriptional and oncogenic mechanisms employed by both VGLL2-NCOA2 and TEAD1-NCOA2 fusion proteins, we expect that the TEAD1-NCOA2 dependent C2C12 transplantation model will closely resemble that induced by VGLL2-NCOA2.

(3) For the fusion-oncogenes, did the HA, FLAG, or V5 tag impact fusion-oncogene activity? Was the tag on the 3' or 5' of the fusion? This was not discussed in the methods.

To address the reviewer’s concern, we carefully compared the transcriptional activity of the fusion proteins with the HA tag at the 5’ end or FLAG and V5 tag at the 3’ end. We found that neither the tag type nor its location significantly affects the ability of VGLL2-NCOA2 and TEAD1-NCOA2 to induce downstream gene transcription, measured by qPCR. The data is summarized in Figure 1-figure supplement 1 G-H.

(4) Generally, the lack of details in the figures, figure legends, and methods make the data difficult to interpret. A few examples are below:a. Individual data points are not shown for figure bar plots (how many technical or biological replicates are present and how many times was the experiment repeated?).

As requested, we have added the individual data points to the bar plots. The Method section now includes information on the number of biological replicates and the times the experiments were repeated.

b. What exons were included in the fusion-oncogenes from VGLL2 and NCOA2 or TEAD1 and NCOA2?

We have now included the exon structure organization of VGLL2-NCOA2 or TEAD1-NCOA2 fusions in Figure 1-figure supplement 1A.

c. For how long were the colony formation experiments performed? Two weeks?

We have included more detailed information about the colony formation assay in the Methods section.

d. In Figure 2D, what concentration of CP1 was used and for how long?

The CP1 concentration and treatment duration information has now been included in the figure legend and Methods section.

e. How was A485 resuspended for cell culture and mouse experiments, what is the percentage of DMSO?

The Methods section now includes detailed information on how A485 is prepared for in vitro and in vivo experiments.

f. How many replicates were done for RNA-seq, CUT&RUN, and ATACseq experiments?

RNA-seq was done with three biological replicates and CUT&RUN and ATAC-seq were performed with two biological replicates. This information is now included in the Methods section for clarification.

**Reviewer #2 (Public Review):**
In the manuscript entitled "VGLL2 and TEAD1 fusion proteins drive YAP/TAZ-independent transcription and tumorigenesis by engaging p300", Gu et al. studied two Hippo pathway-related gene fusion events (i.e., VGLL2-NCOA2, TEAD1-NCOA2) in spindle cell rhabdomyosarcoma (scRMS) and showed that their fusion proteins can activate Hippo downstream gene transcription independent of YAP/TAZ. Using the BioID-based mass spectrometry analysis, the authors revealed histone acetyltransferase CREBBP/EP300 as specific binding proteins for VGLL2-NCOA2 and TEAD1-NCOA2 fusion proteins. Pharmacologically targeting EP300 inhibited the fusion proteins-induced Hippo downstream gene transcription and tumorigenic events.Overall, this study provides mechanistic insights into the scRMS-associated gene fusions in tumorigenesis and reveals potential therapeutic targets for cancer treatment. The manuscript is well-written and easy to follow.Here, several suggestions are made for the authors to improve their study.Main points(1) The authors majorly focused on the Hippo downstream gene transcription in this study, while a significant portion of genes regulated by the VGLL2-NCOA2 and TEAD1-NCOA2 fusion proteins are non-Hippo downstream genes (Figure 3). The authors should investigate whether the altered Hippo pathway transcription is essential for VGLL2-NCOA2 and TEAD1-NCOA2-induced cell transformation and tumorigenesis. Specifically, they should test if treatment with the TEAD inhibitor can reverse the cell transformation and tumorigenesis caused by VGLL2-NCOA2 but not TEAD1-NCOA2. In addition, it is important to examine whether YAP-5SA expression can rescue the inhibitory effects of A485 on VGLL2-NCOA2 and TEAD1-NCOA2-induced colony formation and tumor growth. This will help clarify whether Hippo downstream gene transcription is important for the oncogenic activities of these two fusion proteins.

We thank the reviewer for the comments. Although we have not tested the small molecular TEAD inhibitor on VGLL2-NCOA2 or TEAD1-NCOA2-induced cell transformation and tumorigenesis, we expect that TEAD inhibition will block VGLL2-NCOA2- but not TEAD1-NCOA2-induced oncogenic activity. It is because TEAD1-NCOA2 does not contain the auto-palmitoylation sites and the hydrophobic pocket in the C-terminal YAP-binding domain of TEAD1 that the TEAD small molecule inhibitor occupies (4). We also appreciate the reviewer’s suggestion of YAP5SA rescue experiments. However, due to its strong oncogenic activity, YAP5SA itself can induce robust downstream transcription and cell transformation with or without A485 treatment, as shown in Figure 5. Thus, it will be unlikely to address whether non-Hippo downstream genes induced by the fusions are important for cell transformation and tumorigenesis. Because of the distinct nature of transcriptional and chromatin landscapes controlled by VGLL2-NCOA2/TEAD-NCOA2 and YAP, we speculate that both Hippo and non-Hippo-related downstream genes contribute to the oncogenic activation and tumor phenotypes induced by the fusion proteins.

(2) Rationale for selecting CBP/p300 for functional studies needs to be provided. The BioID-MS experiment identified many interacting proteins for VGLL2-NCOA2 and TEAD1-NCOA2 fusion proteins (Table S4). The authors should explain the scoring system used to identify the high-interacting proteins for VGLL2-NCOA2 and TEAD1-NCOA2 fusion proteins. Was CEP/p300 the top candidates on the list? Providing this information will help justify the focus on CBP/p300 and validate their importance in this study.

We appreciate the reviewer’s point. CBP/P300 is among the top hits in our proteomics screens of both VGLL2-NCOA2 and TEAD1-NCOA2. Our focus on CBP/P300 is mainly due to the well-established interactions between CBP/P300 and the NCOA family transcriptional co-activators, in which the CBP/P300-NCOA complex plays a central role in mediating nuclear receptors-induced transcriptional activation (5). In addition, our data is consistent with another re-current Vgll2 fusion identified in scRMS, VGLL2-CITED2 (6) that has a C-term fusion partner from CITED2, which is a known CBP/P300 interacting protein (7).

(3) p300 was revealed as a key driver for the VGLL2-NCOA2 and TEAD1-NCOA2 fusion proteins-induced transcriptome alteration and tumorigenesis. To strengthen the point, the authors should identify the p300 binding region on VGLL2-NCOA2 and TEAD1-NCOA2 fusion proteins. Mutants with defects in p300 binding/recruitment should be generated and included as a control in the related q-PCR and tumorigenic studies. This work will help confirm the crucial role of p300 in mediating the oncogenic effects of these two fusion proteins.

We thank the reviewer for the suggestion. We have performed the co-immunoprecipitation assay using the deletion mutant form of VGLL2-NCOA2. We have performed additional co-immunoprecipitation experiments and demonstrated that the C-term NCOA2 part of the fusion is responsible for mediating the interaction between the fusion protein and CBP/P300. These results are now included in the new Figure 5A and are consistent with the reported structural analysis of CBP/P300-NCOA complex (8). In addition, our new data showed the inability of the VGLL2-NCOA2 ∆NCOA2 mutant to induce gene transcription (Figure 1-figure supplement 1D). Furthermore, our data using the small molecular CBP/P300 inhibitor clearly demonstrated that CBP/P300 is required to mediate cell transformation and tumorigenesis induced by the two fusion proteins in vitro and in vivo (Figure 5 and 6).

(4) Another major issue is the overexpression system extensively used in this study. It is important to determine whether the VGLL2-NCOA2 and TEAD1-NCOA2 fusion genes are also amplified in cancer. If not, the expression levels of the VGLL2-NCOA2 and TEAD1-NCOA2 fusion proteins should be adjusted to endogenous levels to assess their oncogenic effects on gene transcription and tumorigenesis. This approach would make the study more relevant to the pathological conditions observed in scRMS cancer patients.

We appreciate the reviewer’s input and acknowledge the limitation of the HEK293T and C2C12 cell-based models that rely on ectopic expression of VGLL2-NCOA2 and TEAD1-NCOA2 fusion proteins. It is currently unclear whether the VGLL2-NCOA2 and TEAD1-NCOA2 fusion genes are also amplified in sarcoma. As mentioned before, these surrogate cell culture systems allowed us to systemically compare the transcriptional regulation by the fusion proteins and YAP/TAZ and elucidate the molecular mechanism underlying the Hippo/YAP-independent oncogenic transformation induced by VGLL2-NCOA2 and TEAD1-NCOA2.

References:

(1) Genes Dev . 2007 Nov 1;21(21):2747-61. doi: 10.1101/gad.1602907. Inactivation of YAP oncoprotein by the Hippo pathway is involved in cell contact inhibition and tissue growth control

(2) Genes Dev . 2010 Jan 1;24(1):72-85. doi: 10.1101/gad.1843810. A coordinated phosphorylation by Lats and CK1 regulates YAP stability through SCF(beta-TRCP)

(3) VGLL2-NCOA2 leverages developmental programs for pediatric sarcomagenesis. Watson S, LaVigne CA, Xu L, Surdez D, Cyrta J, Calderon D, Cannon MV, Kent MR, Cell Rep. 2023 Jan 31;42(1):112013.

(4) Lats1/2 Sustain Intestinal Stem Cells and Wnt Activation through TEAD-Dependent and Independent Transcription. Cell Stem Cell. 2020 May 7;26(5):675-692.e8.

(5) Yi, P., Yu, X., Wang, Z., and O’Malley, B.W. (2021). Steroid receptor-coregulator transcriptional complexes: new insights from CryoEM. Essays Biochem. 65, 857–866.

(6) A Molecular Study of Pediatric Spindle and Sclerosing Rhabdomyosarcoma: Identification of Novel and Recurrent VGLL2-related Fusions in Infantile Cases. Am J Surg Pathol . 2016 Feb;40(2):224-35. doi: 10.1097/

(7) CITED2 and the modulation of the hypoxic response in cancer. Fernandes MT, Calado SM, Mendes-Silva L, Bragança J.World J Clin Oncol. 2020 May 24;11(5):260-274.

(8) Yu, X., Yi, P., Hamilton, R.A., Shen, H., Chen, M., Foulds, C.E., Mancini, M.A., Ludtke, S.J., Wang, Z., and O’Malley, B.W. (2020). Structural insights of transcriptionally active, full-length Androgen receptor coactivator complexes. Mol. Cell 79, 812–823.e4.